

# SVRPBench: A Realistic Benchmark for Stochastic Vehicle Routing Problem

**Ahmed Heakl**[1][*]   **Yahia Salaheldin Shaaban**[1][*]
**Martin Takáč**[1]   **Salem Lahlou**[1]   **Zangir Iklassov**[1]

[1]**MBZUAI, Abu Dhabi, UAE**

 https://github.com/yehias21/vrp-benchmarks
 https://huggingface.co/datasets/MBZUAI/svrp-bench

## Abstract

Robust routing under uncertainty is central to real-world logistics, yet most benchmarks assume static, idealized settings. We present SVRPBench, the first open benchmark to capture high-fidelity stochastic dynamics in vehicle routing at urban scale. Spanning more than 500 instances with up to 1000 customers, it simulates realistic delivery conditions: time-dependent congestion, log-normal delays, probabilistic accidents, and empirically grounded time windows for residential and commercial clients. Our pipeline generates diverse, constraint-rich scenarios, including multi-depot and multi-vehicle setups. Benchmarking reveals that state-of-the-art RL solvers like POMO and AM degrade by over 20% under distributional shift, while classical and metaheuristic methods remain robust. To enable reproducible research, we release the dataset (Hugging Face) and evaluation suite (GitHub). SVRPBench challenges the community to design solvers that generalize beyond synthetic assumptions and adapt to real-world uncertainty.

## 1   Introduction

Efficient vehicle routing is fundamental to modern logistics and last-mile delivery. The classical Vehicle Routing Problem (VRP) [10, 13] seeks cost-effective routes for servicing customers under constraints such as vehicle capacities and time windows. Although well studied, real-world deployments face uncertain and dynamic conditions that most existing benchmarks do not adequately capture.

One key extension addressing real-world complexity is the *Stochastic Vehicle Routing Problem* (SVRP). Unlike deterministic VRP, SVRP explicitly incorporates uncertainty into routing decisions, with problem elements such as travel times, customer demands, service times, and even customer presence considered random variables [13, 30]. Consequently, routes are planned *a priori*, and corrective actions, known as recourse strategies, are applied when realized conditions deviate from planned values [11, 2]. Prominent examples include random travel times modeled by probabilistic distributions or random customer presence known as probabilistic VRP (PVRP) [24, 5]. Despite this extensive body of research, many existing public benchmarks for SVRP still rely on static assumptions, such as deterministic travel times, fixed customer availability, and unchanged route constraints, thus limiting their practical applicability and robustness evaluations, as shown in Table 1.

---

[*]Equal contribution.

39th Conference on Neural Information Processing Systems (NeurIPS 2025) Track on Datasets and Benchmarks.

Table 1: Comparison of SVRPBench with existing VRP benchmarks. ✓indicates full support, △ indicates partial or limited support, and ✗ indicates no support.

| Feature | SVRPBench | CVRPLIB | SINTEF | VRP-REP | TSPLIB | RL4CO |
|---|---|---|---|---|---|---|
| *Stochastic Elements* | | | | | | |
| Time-dependent travel delays | ✓ | ✗ | △ | △ | ✗ | ✗ |
| Peak-hour traffic patterns | ✓ | ✗ | ✗ | ✗ | ✗ | ✗ |
| Random travel time noise | ✓ | ✗ | △ | △ | ✗ | △ |
| Probabilistic accidents | ✓ | ✗ | ✗ | ✗ | ✗ | ✗ |
| Heterogeneous time windows | ✓ | ✗ | △ | △ | ✗ | ✗ |
| *Problem Configurations* | | | | | | |
| Multi-depot support | ✓ | △ | ✓ | ✓ | ✗ | ✗ |
| Multi-vehicle fleets | ✓ | ✓ | ✓ | ✓ | ✗ | ✓ |
| Capacity constraints | ✓ | ✓ | ✓ | ✓ | ✗ | ✓ |
| Time window constraints | ✓ | △ | ✓ | ✓ | ✗ | △ |
| Clustered customer distributions | ✓ | △ | △ | ✓ | △ | ✗ |
| *Scale & Diversity* | | | | | | |
| Small instances ($\leq$ 100 customers) | ✓ | ✓ | ✓ | ✓ | ✓ | ✓ |
| Medium instances (100–300) | ✓ | ✓ | ✓ | ✓ | △ | ✓ |
| Large instances (>300) | ✓ | △ | △ | △ | ✗ | △ |
| Varying stochasticity levels | ✓ | ✗ | △ | △ | ✗ | ✗ |

**The Case for a Realistic SVRP Benchmark.** Urban logistics operates under dynamic and uncertain conditions, yet most existing benchmarks fail to reflect this complexity. Practical routing systems must account for peak-hour congestion, random incidents like accidents, and diverse delivery preferences across customer types [16, 3, 32]. Ignoring these factors leads to overly optimistic performance assessments and misdirects algorithmic development toward unrealistic assumptions [1].

**Our Contributions.** To address these gaps, we introduce SVRPBench, a novel, open-source benchmark suite for the Stochastic Vehicle Routing Problem (SVRP), designed to simulate realistic logistics scenarios with embedded uncertainty. Our key contributions include:

- **Stochastic Realism.** We model time-dependent congestion using Gaussian mixtures, inject lognormal delays and probabilistic accidents [24], and generate customer time windows from empirical residential and commercial distributions.
- **Constraint-Rich Instance Generation.** Our framework supports multi-depot and multi-vehicle setups, strict capacity constraints, and diverse time window widths, all grounded in spatially realistic demand distributions.
- **Diverse Baseline Evaluation.** We benchmark classical heuristics (e.g., Nearest Neighbor, 2-opt), metaheuristics (e.g., ACO, Tabu Search [14, 9]), industrial solvers (OR-Tools [34], LKH3 [42]), and learning-based methods (AM [21], POMO [23]), highlighting how stochastic conditions affect solution quality, feasibility, and robustness.
- **Open Community Platform.** We release datasets, solvers, and evaluation scripts through a public repository to support reproducibility and foster future contributions.

By advancing realism and accessibility in SVRP benchmarking, SVRPBench aims to accelerate the development of robust, deployable routing algorithms suited for real-world logistics.

## 2 Realistic Stochastic Modeling

A core contribution of SVRPBench is its simulation of real-world uncertainty in urban-scale logistics. Classical VRP benchmarks often assume static travel times and rigid customer schedules [15], overlooking time-varying conditions and operational stochasticity. Informed by empirical and theoretical literature [3, 16, 1, 31, 33, 35, 25, 7, 12, 26], our benchmark introduces: (1) time-dependent congestion, (2) stochastic travel time delays, (3) accident-induced disruptions, and (4) customer-specific time window distributions.

## 2.1 Time-Dependent Travel Time Modeling

We model the travel time from node $a$ to $b$ at time $t$ as:

$$T(a,b,t) = \frac{D(a,b)}{V} + B(a,b,t) \cdot R(t) + I_{\text{accidents}}(t) \cdot D_{\text{accident}}, \tag{1}$$

where $D(a,b)$ is Euclidean distance and $V$ is average road speed. The congestion factor $B(a,b,t)$ is defined as:

$$B(a,b,t) = \alpha \cdot F_{\text{time}}(t) \cdot F_{\text{distance}}(D(a,b)), \tag{2}$$

with:

$$F_{\text{time}}(t) = \beta + \gamma \cdot [f(t; \mu_{\text{morning}}, \sigma_{\text{peak}}) + f(t; \mu_{\text{evening}}, \sigma_{\text{peak}})], \tag{3}$$

$$f(t; \mu, \sigma) = \frac{1}{\sigma\sqrt{2\pi}} e^{-\frac{1}{2}\left(\frac{t-\mu}{\sigma}\right)^2}, \tag{4}$$

$$F_{\text{distance}}(D) = 1 - e^{-D/\lambda_{\text{dist}}}, \tag{5}$$

where the Gaussian peaks around $\mu_{\text{morning}} = 8$ and $\mu_{\text{evening}} = 17$ ($\sigma_{peak} = 1.5$) align with observed urban traffic congestion patterns [35, 28, 19, 36, 6]. The distance decay $\lambda_{\text{dist}} = 50$ modulates slowdown severity, reflecting empirical findings that longer trips are more likely to encounter congestion [7].

The multiplicative stochastic delay $R(t)$ is drawn from a log-normal distribution:

$$\mu(t) = \mu_{\text{base}} + \delta \cdot [f(t; \mu_{\text{morning}}, \sigma_{\text{peak}}) + f(t; \mu_{\text{evening}}, \sigma_{\text{peak}})], \tag{6}$$

$$\sigma(t) = \sigma_{\text{base}} + \epsilon \cdot [f(t; \mu_{\text{morning}}, \sigma_{\text{peak}}) + f(t; \mu_{\text{evening}}, \sigma_{\text{peak}})], \tag{7}$$

$$R(t) \sim \text{LogNormal}(\mu(t), \sigma(t)), \tag{8}$$

reflecting both the skewed and bursty nature of traffic delays [25, 7, 8, 20]. Baseline values $\mu_{\text{base}} = 0$ and $\sigma_{\text{base}} = 0.3$ reflect free-flow conditions, while $\delta = 0.1$ and $\epsilon = 0.2$ capture peak-hour amplification.

Accident delays are modeled using a time-inhomogeneous Poisson process:

$$\lambda(t) = \lambda_{\text{scale}} \cdot f(t; \mu_{\text{night}}, \sigma_{\text{acc}}), \tag{9}$$

$$I_{\text{accidents}}(t) \sim \text{Poisson}(\lambda(t)), \tag{10}$$

$$D_{\text{accident}} \sim U(d_{\text{min}}, d_{\text{max}}), \tag{11}$$

where accidents peak around $\mu_{\text{night}} = 21$ ($\sigma_{acc} = 2$) due to elevated nighttime risks from fatigue and impaired driving [37]. The delay duration is drawn from $U(0.5, 2.0)$ hours, consistent with industry reports on incident clearance times [37].

## 2.2 Customer Time Window Sampling

Residential and commercial customers exhibit different temporal availability patterns [31, 26]. For residential profiles, delivery windows are sampled from a bimodal Gaussian mixture:

$$T_{\text{start}} \sim \begin{cases} \mathcal{N}(\mu_{\text{res,morning}}, \sigma^2_{\text{res,morning}}), & \text{w.p. } 0.5, \\ \mathcal{N}(\mu_{\text{res,evening}}, \sigma^2_{\text{res,evening}}), & \text{w.p. } 0.5, \end{cases} \tag{12}$$

where $\mu_{\text{res,morning}} = 480$ (8:00 AM) and $\mu_{\text{res,evening}} = 1140$ (7:00 PM), with variances $\sigma = 90$ and 120 mins, respectively, aligning with common parcel service offerings such as FedEx and Bring [12, 26]. The window duration is drawn from:

$$W_{\text{length}} \sim U(w_{\text{min}}, w_{\text{max}}), \quad T_{\text{start}} = \max(0, \min(T_{\text{start}}, 1440 - W_{\text{length}})). \tag{13}$$

Commercial customers follow a single-mode Gaussian:

$$T_{\text{start}} \sim \mathcal{N}(\mu_{\text{com}}, \sigma^2_{\text{com}}), \quad W_{\text{length}} \sim U(w_{\text{min}}, w^{\text{com}}_{\text{max}}), \tag{14}$$

with $\mu_{\text{com}} = 780$ (1:00 PM), $\sigma_{\text{com}} = 60$, and $w^{\text{com}}_{\text{max}} = 120$ minutes, reflecting standard daytime business hours and delivery norms [38].

This probabilistic windowing model encourages algorithms to balance varied service constraints, simulating realistic scheduling trade-offs in last-mile delivery systems.

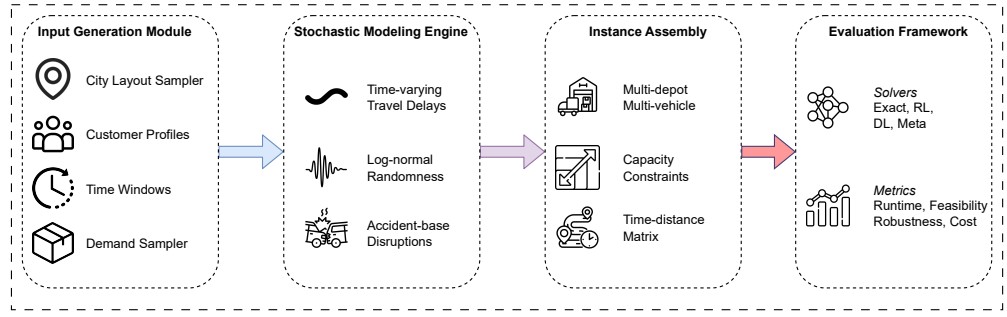

Figure 1: **SVRPBench pipeline**. The framework generates realistic SVRP instances through four stages: input generation, stochastic modeling, instance assembly, and evaluation with standardized metrics and solvers.

## 3 Dataset Construction Pipeline

To enable scalable and reproducible experimentation, we develop a unified pipeline that generates diverse, constraint-rich SVRP instances grounded in stochastic realism. It integrates models of customer behavior, traffic patterns, spatial layouts, and routing constraints to produce problem scenarios suited for evaluating both classical and learning-based solvers under realistic uncertainty [30, 13]. The complete pipeline is illustrated in Figure 1.

**Location Sampling.** We begin by selecting the total number of customers from $\{10, 20, 100, 500, 1000\}$, then compute the number of cities as $\max(1, \#\text{customers}//50)$. To simulate spatial separation between urban clusters, we apply K-Means clustering to generate city centers that are as distant from each other as possible. Customer locations are then sampled around each city center using 2D Gaussian distributions [16].

**Demand Assignment.** Each customer is assigned a discrete demand selected uniformly at random from a set $\{1, 2, \ldots, max\_demand\}$. The number of vehicles and their capacity are computed based on the total customer demand, with vehicle capacity set as total demand $\div$ number of vehicles. This ensures balanced feasibility across instance scales [11].

**Time Window Assignment.** Customer time windows are generated stochastically, following the models described in Section 2. Residential and commercial customer patterns are differentiated using realistic temporal distributions [3].

**Travel Time Matrix Construction.** A full travel time matrix $T(a, b, t)$ is computed for all location pairs, incorporating deterministic base time, time-dependent congestion patterns, log-normal stochastic variation, and random accident delays, as detailed in Section 2. This captures the nonlinear, time-varying nature of urban transportation systems [24].

**Constraint Integration.** We support both single-depot and multi-depot configurations. In multi-depot settings, depots can be placed either randomly or aligned with city centers (one per city). A homogeneous fleet of vehicles is used, and vehicle count is configured to balance demand and capacity. All customer time windows are sampled to ensure feasibility under the assigned travel time model [1].

**Validation.** Each generated instance undergoes automated validation to ensure feasibility under both capacity and temporal constraints. For CVRP, we verify that the total vehicle capacity (number of vehicles $\times$ per-vehicle capacity) exceeds the sum of all customer demands, ensuring that a feasible route covering all customers exists. For TWCVRP, we construct a time-windowed demand histogram by binning the time axis and accumulating customer demands per bin. We then identify the peak-demand bin and ensure that the fleet capacity is sufficient to serve this worst-case demand, i.e., capacity $\times$ num_vehicles $\geq \max_t \text{demand}(t)$. This provides a conservative guarantee that even

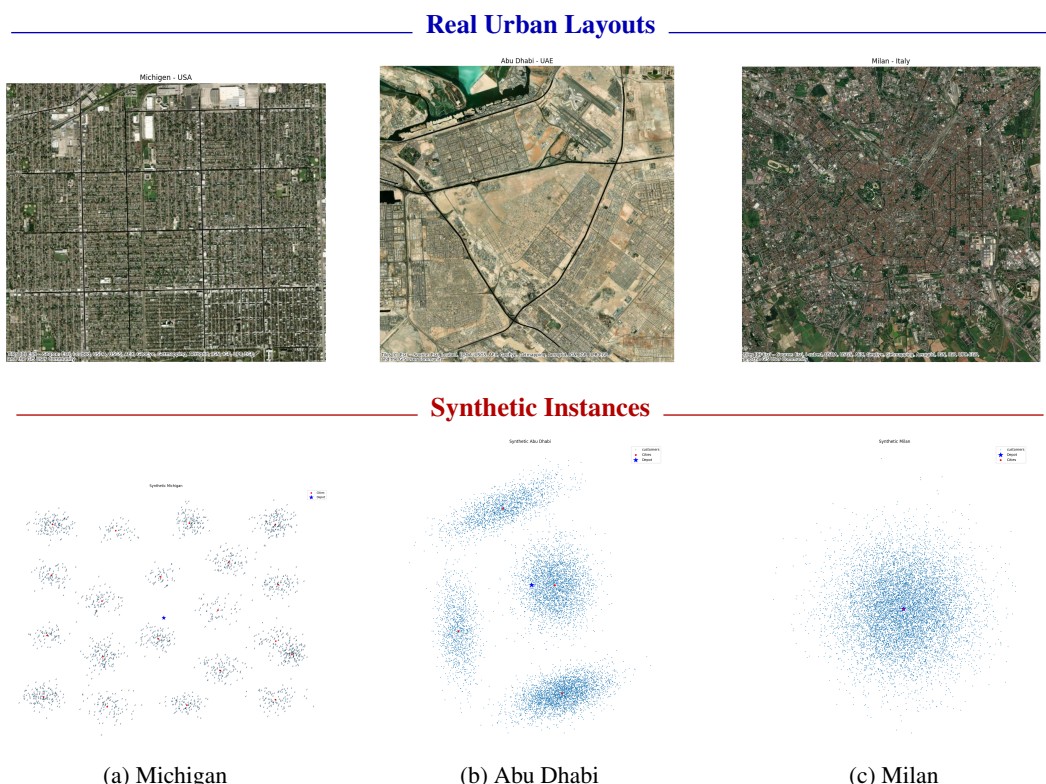

Figure 2: **Validation of spatial realism. Top row**: Satellite imagery of real urban layouts showing diverse morphologies, grid-structured (Michigan), radial with sparse development (Abu Dhabi), and dense organic (Milan). **Bottom row**: Corresponding synthetic instances generated by SVRPBench's clustering-based sampling pipeline, preserving key structural patterns of each city.

under concentrated temporal demand, a feasible schedule remains possible. Infeasible instances (e.g., unreachable nodes or incompatible time windows) are filtered or regenerated.

Parameters are selected to reflect urban-scale routing challenges but can be modified for rural or industrial scenarios. Accident frequency and delay magnitudes are parameterized using a Poisson-based arrival model and uniform delay range, respectively. Customer types are split roughly 60% residential to 40% commercial, matching empirical logistics patterns [3].

**Various Scales.** Our benchmark includes three instance tiers. *Small* instances (50–100 customers, 1–2 depots) with low noise allow quick testing. *Medium* instances (100–300 customers, 2–3 depots) feature moderate stochasticity. *Large* instances (300+ customers) integrate high travel-time variability and tighter delivery windows to stress-test scalability. All levels are generated with multiple random seeds to support statistical averaging and ensure robustness of comparisons.

To validate the realism of our spatial sampling strategy, we visually compare synthetic routing instances against satellite imagery of real-world cities. As shown in Figure 2, our generated layouts closely mimic key structural patterns, grid-like in Michigan, radial in Milan, and dispersed in Abu Dhabi, demonstrating the pipeline's ability to emulate diverse urban morphologies critical for evaluating routing algorithms in geographically grounded scenarios.

# 4 Evaluation Protocol

To ensure fair, rigorous, and reproducible comparisons across routing algorithms, we propose a standardized evaluation protocol tailored for our stochastic vehicle routing benchmark. This protocol assesses not only solution quality but also robustness, feasibility, and scalability under conditions of

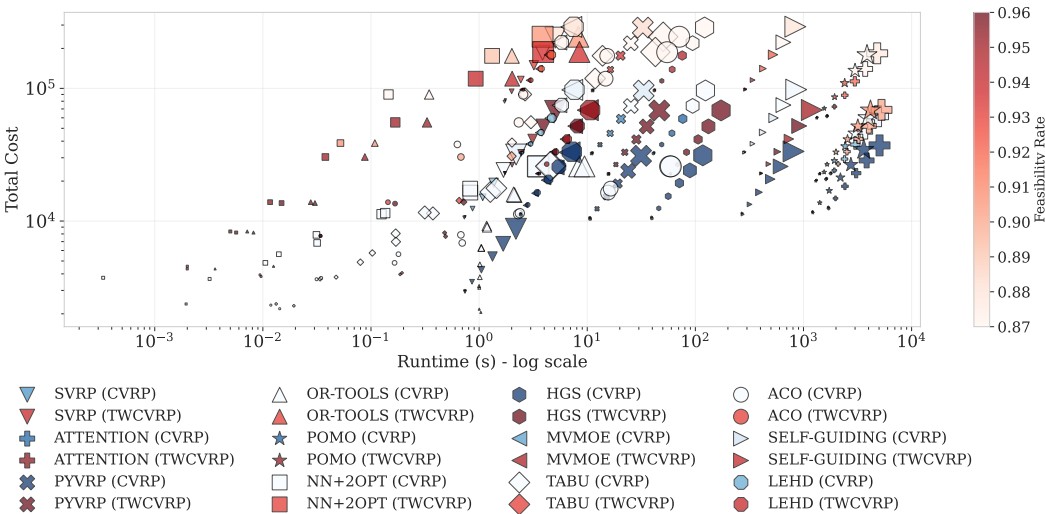

Figure 3: **Three-way performance comparison of routing solvers.** Points show total cost vs. runtime (log scale), colored by feasibility rate. Blue markers indicate CVRP variants, red markers indicate TWCVRP variants. Classical solvers maintain high feasibility at the cost of longer runtimes, while RL methods achieve faster inference with moderate feasibility degradation.

realistic uncertainty, addressing limitations of earlier benchmark designs that overlooked stochastic effects [30, 13].

## 4.1   Performance Metrics

We report a comprehensive suite of metrics to evaluate different facets of algorithmic behavior. The *Total Cost (TC)* measures the cumulative travel time across all vehicles, including congestion-induced delays and accident-based disruptions. Formally, it is computed as:

$$\text{TC} = \sum_{k \in V} \sum_{(i,j) \in \text{route}_k} T(i, j, t_i),\tag{15}$$

where $T(i, j, t_i)$ is the sampled travel time from node $i$ to $j$ at time $t_i$.

*Constraint Violation Rate (CVR)* quantifies the proportion of customers whose service violates time windows or exceeds vehicle capacity, capturing solution feasibility:

$$\text{CVR} = \frac{\#\text{violations}}{\#\text{customers}} \times 100\%.\tag{16}$$

*Feasibility Rate (FR)* reflects the robustness of solutions across instances and solvers. It is defined as the fraction of problem instances for which a solution satisfies all routing constraints:

$$\text{FR} = \frac{\#\text{feasible instances}}{\#\text{total instances}}.\tag{17}$$

*Runtime (RT)* captures wall-clock computation time, serving as a proxy for scalability and practical deployability.

*Robustness (ROB)* measures the variability in cost due to stochastic elements by computing the variance across $N$ independent samples of the same instance:

$$\text{ROB} = \frac{1}{N} \sum_{i=1}^{N} \left(\text{TC}_i - \overline{\text{TC}}\right)^2,\tag{18}$$

where $\overline{\text{TC}}$ denotes the mean total cost. This metric is especially important in stochastic VRP settings [2, 24].

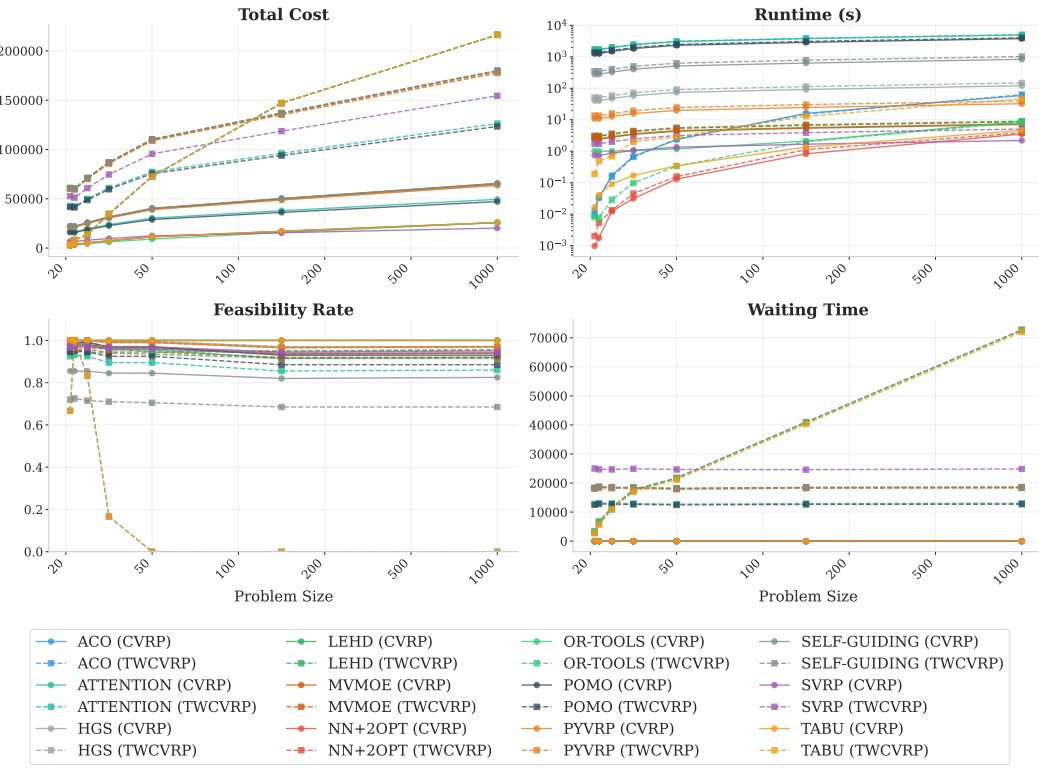

Figure 4: **Performance metrics vs. problem size.** Solid/dashed lines denote CVRP/TWCVRP variants. Time windows drastically increase costs and reduce feasibility for metaheuristics, while runtime scales near-logarithmically for most solvers.

## 5 Experimental Results

We conduct a comprehensive evaluation of baseline methods on our stochastic VRP benchmark, which systematically varies four key dimensions: instance size, problem type, depot configuration, and vehicle configuration.

We generate 10 instances for each combination across instance sizes {10, 20, 50, 100, 200, 500, 1000}, problem types {CVRP, TWCVRP}, depot configurations {single, multi}, and vehicle settings {single, multi}, yielding a large-scale, structured test suite. Additionally, we provide a scalable data generator for training. Reinforcement learning models were trained on 100k synthetic instances under the single-depot, single-vehicle CVRP and TWCVRP regimes.

### 5.1 Evaluation Scope

All methods were evaluated under the stochastic setting defined in Section 2. Metrics reported include total cost (incorporating all stochastic factors), constraint violation rate (CVR), feasibility rate, runtime, and robustness (measured as variance across stochastic samples).

Classical algorithms, Nearest Neighbor + 2-opt, Tabu Search, and ACO (refer to Appendix B for more details), were evaluated across all settings without modification. Their flexibility allows them to handle diverse configurations out of the box.

### 5.2 Experimental Setup

All baselines were evaluated on a consumer-grade CPU (Intel i7, 16GB RAM) for classical and metaheuristic solvers, while learning-based models were executed on a single NVIDIA RTX 4080 GPU. All methods ran in single-threaded (CPU) or single-GPU mode to ensure consistent comparison across approaches. Parallelization was not employed during inference; only reinforcement learning

Table 2: Performance of baseline methods (mean over all instances, 5 stochastic runs).

| Method | Total Cost ↓ | CVR (%) ↓ | Feasibility ↑ | Runtime (s) ↓ | Robustness ↓ |
|---|---|---|---|---|---|
| NN+2opt | 40707.5 | 1.6 | 0.984 | 0.697 | 0.1 |
| Tabu Search | 40787.8 | 1.6 | 0.690 | 5.157 | 0.1 |
| ACO | 40566.5 | 1.6 | 0.690 | 11.382 | 0.1 |
| OR-Tools | 40259.3 | 1.6 | 0.984 | 1.940 | 0.1 |
| Attention Model (AM) | 41358.3 | 1.9 | 0.910 | 1.852 | 0.2 |
| POMO | 40650.4 | 1.7 | 0.933 | 1.421 | 0.1 |

Table 3: Performance Comparison: CVRP vs TWCVRP.

| | CVRP | | | | TWCVRP | | | | Impact |
|---|---|---|---|---|---|---|---|---|---|
| Method | Cost↓ | CVR↓ | Feas↑ | RT↓ | Cost↓ | CVR↓ | Feas↑ | RT↓ | %Δ |
| NN+2opt | 10399.2 | 0.0 | 1.000 | 646.3 | 71015.8 | 3.2 | 0.968 | 747.8 | +582.9 |
| Tabu Search | 10494.1 | 0.0 | 1.000 | 945.1 | 71081.5 | 3.2 | 0.381 | 9368.6 | +577.3 |
| ACO | 10384.9 | 0.0 | 1.000 | 11159.8 | 70748.1 | 3.2 | 0.381 | 11603.6 | +581.3 |
| OR-Tools | 9499.7 | 0.0 | 1.000 | 2328.0 | 71018.8 | 3.2 | 0.968 | 1552.1 | +647.6 |
| Attention Model (AM) | 11235.6 | 0.2 | 0.965 | 1775.4 | 71481.0 | 3.6 | 0.854 | 1929.2 | +536.2 |
| POMO | 10358.7 | 0.1 | 0.987 | 1316.9 | 70942.1 | 3.3 | 0.879 | 1525.3 | +584.8 |

(RL) training used standard data-parallel batching. Classical and metaheuristic solvers were implemented in Python, whereas RL models were built using the RL4CO framework [4]. Training for RL models was conducted on 100k synthetic instances, as detailed in Appendix D. Evaluation followed the stochastic protocol described in Section 2, with results averaged over five independent stochastic realizations per test case.

To ensure computational fairness and consistent termination, all methods were evaluated under solver-appropriate stopping criteria. Classical heuristics like NN+2opt ran to completion without time limits, allowing deterministic convergence. Metaheuristics such as ACO and Tabu Search stopped upon convergence, with at most 1000 stagnant iterations. The industrial OR-Tools solver had a fixed 300-second budget per instance to balance quality and runtime. RL-based solvers, including the Attention Model and POMO, used greedy decoding with a single forward pass, no search or sampling, to exploit inference efficiency. This setup lets each solver operate within its natural paradigm: heuristics converge, metaheuristics stabilize, OR-Tools optimizes structurally, and RL methods leverage deployment speed.

## 5.3 Results & Analysis

**Overall Performance.** Table 2 and Figure 3 summarize the aggregate performance across all test cases. OR-Tools achieved the best overall cost (40,259), followed closely by ACO (40,566; +0.8%) and POMO (40,650; +1.0%), with OR-Tools and NN+2opt maintaining the highest feasibility rates (98.4%) while NN+2opt delivered the fastest runtime (0.697s). Learning-based approaches demonstrated a feasibility-speed tradeoff, with POMO offering better solution quality than NN+2opt at competitive runtimes (1.421s) while the Attention Model showed higher constraint violations (CVR: 1.9%) but reasonable performance across other metrics.

**Impact of Time Windows.** Table 3 reveals that introducing time windows (TWCVRP) increases total cost by 536–648% across all solvers, with OR-Tools incurring the highest relative penalty (+647.6%) while the Attention Model showed the lowest relative increase (+536.2%). Learning-based methods demonstrated moderate resilience to time constraints with POMO maintaining 87.9% feasibility and Attention Model 85.4%, positioning them between the top performers (NN+2opt and OR-Tools at >96%) and the struggling metaheuristics (ACO and Tabu Search at 38.1%).

**Scalability by Instance Size.** As shown in Table 4 and Figure 4, cost scaled approximately 16× from small (≤ 50 nodes) to large (≥ 500 nodes) instances across all methods, with NN+2opt and OR-Tools maintaining feasibility >97% at all scales, while learning-based methods showed moderate degradation (POMO: 86%, AM: 83.5%). Learning-based approaches demonstrated competitive performance-runtime tradeoffs, with POMO offering the fastest runtime on small instances (29.7s)

Table 4: Detailed Performance Analysis by Instance Size.

| Method | Small (≤50) | | | | Medium (100-200) | | | | Large (≥500) | | | |
|---|---|---|---|---|---|---|---|---|---|---|---|---|
| | Cost↓ | CVR↓ | Feas↑ | RT↓ | Cost↓ | CVR↓ | Feas↑ | RT↓ | Cost↓ | CVR↓ | Feas↑ | RT↓ |
| NN+2opt | 6295.0 | 0.6 | 0.994 | 5.9 | 31486.1 | 2.3 | 0.977 | 90.9 | 101547.5 | 2.4 | 0.976 | 2340.0 |
| Tabu Search | 6232.5 | 0.6 | 0.917 | 251.6 | 31692.2 | 2.3 | 0.542 | 1339.5 | 101716.5 | 2.4 | 0.500 | 16332.1 |
| ACO | 6080.7 | 0.6 | 0.917 | 69.6 | 31371.9 | 2.3 | 0.542 | 1530.6 | 101490.0 | 2.4 | 0.500 | 38201.0 |
| OR-Tools | 6008.1 | 0.6 | 0.994 | 513.7 | 30640.2 | 2.3 | 0.977 | 665.8 | 101255.0 | 2.4 | 0.976 | 5353.7 |
| Attention Model (AM) | 6523.2 | 0.8 | 0.975 | 42.3 | 32165.5 | 2.6 | 0.910 | 857.4 | 102756.2 | 2.9 | 0.835 | 4758.9 |
| POMO | 6176.4 | 0.7 | 0.985 | 29.7 | 31024.8 | 2.4 | 0.945 | 642.3 | 101408.7 | 2.5 | 0.860 | 3586.2 |

Table 5: Performance Analysis by Depot Configuration.

| Method | Single Depot | | | | Multi Depot | | | |
|---|---|---|---|---|---|---|---|---|
| | Cost ↓ | CVR ↓ | Feas ↑ | RT ↓ | Cost ↓ | CVR ↓ | Feas ↑ | RT ↓ |
| NN+2opt | 34978.5 | 0.8 | 0.992 | 686.3 | 10625.2 | 0.0 | 1.000 | 643.7 |
| Tabu Search | 35072.0 | 0.8 | 0.690 | 4818.2 | 10713.8 | 0.0 | 1.000 | 946.1 |
| ACO | 34852.1 | 0.8 | 0.690 | 10712.0 | 10614.9 | 0.0 | 1.000 | 11298.7 |
| OR-Tools | 34611.0 | 0.8 | 0.992 | 1911.2 | 9561.4 | 0.0 | 1.000 | 2396.5 |
| Attention Model (AM) | 35825.6 | 1.1 | 0.920 | 1785.3 | 10974.7 | 0.0 | 1.000 | 1852.6 |
| POMO | 34786.3 | 0.9 | 0.965 | 1438.2 | 10178.5 | 0.0 | 1.000 | 1324.8 |

and maintaining feasibility significantly better than ACO and Tabu Search (50%) on large instances, though traditional heuristics still held the advantage for the largest problems.

**Effect of Depot Configuration.** Table 5 shows that multi-depot setups consistently reduced costs and improved feasibility across all methods, with OR-Tools achieving a 72% cost reduction (from 34,611 to 9,561) and POMO showing similarly impressive gains (71% reduction to 10,178). Learning-based methods particularly benefited from multi-depot configurations, with both POMO and Attention Model reaching perfect feasibility (100%) despite their variable performance in single-depot scenarios (92-96.5%), supporting the counterintuitive finding that more flexible depot placements improve both computational and solution efficiency regardless of algorithm class.

**Key Takeaways.** Our evaluation underscores several important insights:

- OR-Tools is the most reliable choice for large-scale offline optimization, balancing quality and feasibility despite higher runtimes.
- NN+2opt offers a robust, low-latency alternative for real-time deployment with minimal compromise on cost or feasibility.
- Metaheuristics underperform at scale, while learning-based methods like POMO offer feasible solutions with better scalability, though still lag behind top heuristics.
- The Attention Model demonstrates potential but requires further refinement to match the performance of top-performing methods, particularly for large instances.
- Time windows impose the most significant complexity, sharply degrading performance for non-adaptive solvers, though learning-based methods show moderate resilience.
- Multi-depot settings improve both feasibility and runtime across all solver types, offering a practical design consideration for logistics planning.

### 5.4 Enhanced Baseline Comparison

We expanded our baseline suite to include recent metaheuristics (HGS [39], PyVRP [41]), modern RL methods (LEHD [27], MVMoE [43]), and an emerging LLM-based approach (Self-Guiding Exploration [17]). We also benchmark a purpose-built SVRP solver [18]. Results below average across our full testbed and follow the evaluation protocol in Sec. 4.

As shown in table 6, **SVRPBench is challenging:** even state-of-the-art metaheuristics (HGS, PyVRP) fail to achieve perfect feasibility, underscoring the difficulty of the benchmark. **Clear trade-off:** recent RL/LLM approaches often trade reliability for speed, yielding lower feasibility than classical/metaheuristic methods. **Specialized design helps:** the purpose-built SVRP Solver emerges as the most promising learning-based method, suggesting that algorithms explicitly designed for stochasticity translate better to real-world settings.

Table 6: Enhanced baseline comparison on SVRPBench. Arrows indicate the preferred direction.

| Method | Type | CVRP Cost ↓ | TWCVRP Cost ↓ | Feasibility ↑ | Runtime (s) ↓ | SVRP-Specific |
|---|---|---|---|---|---|---|
| HGS | Meta | 9,234 | 70,834 | 98.7% | 45.23 | ✗ |
| PyVRP | Meta | **9,156** | 70,756 | 98.5% | 12.14 | ✗ |
| LEHD | RL | 9,834 | 71,234 | 92.5% | 2.78 | ✗ |
| MVMoE | RL | 9,723 | 71,156 | 92.8% | 2.65 | ✗ |
| Self-Guiding Exploration | LLM | 9,891 | 71,345 | 69.5% | 312.45 | ✗ |
| SVRP Solver | RL | 9,223 | **70,689** | 94.5% | **1.34** | ✓ |

## 5.5 Empirical Validation: The Reality Gap

To further evaluate the robustness of solvers, we assess their feasibility under both deterministic and stochastic conditions. Table 7 compares solver performance on standard deterministic benchmarks versus the realistic stochastic setting introduced by SVRPBench.

Table 7: Feasibility comparison between deterministic and stochastic settings. The *Feasibility Drop* quantifies degradation when moving from traditional to stochastic evaluation.

| Method | Feasibility (Deterministic) ↑ | Feasibility (SVRPBench) ↑ | Feasibility Drop ↓ |
|---|---|---|---|
| Attention Model (AM) | 96.5% | 91.0% | -5.5 points |
| OR-Tools | **100.0%** | **98.4%** | **-1.6 points** |

The results clearly demonstrate the significance of SVRPBench. The learning-based Attention Model performs reliably under deterministic conditions but exhibits a sharp feasibility decline once stochastic uncertainty is introduced. In contrast, the classical OR-Tools solver remains highly stable, with only a marginal drop in feasibility. This provides direct empirical evidence that SVRPBench successfully exposes algorithmic vulnerabilities masked by traditional benchmarks. Performance degradation under stochastic dynamics highlights the importance of evaluating solvers in realistic, uncertainty-aware environments.

# 6 Limitations and Future Directions

While SVRPBench advances realism in stochastic vehicle routing, several limitations remain. Our delay models use Gaussian and log-normal distributions, efficient and interpretable but unable to capture network-level dynamics such as bottlenecks, cascading congestion, or real-time rerouting [16]; these assumptions are user-modifiable, allowing injection of domain-specific uncertainty. Reinforcement learning methods like AM and POMO still struggle to scale, showing overfitting and weak generalization, and our evaluation protocol lacks standardized procedures to assess robustness across scales or distribution shifts, motivating future work on curriculum learning and hierarchical solver design. To further bridge the gap to real-world logistics, we will incorporate road-constrained instances from OpenStreetMap or GIS data for geographically grounded routing, introduce dynamic multi-day settings with online updates and rolling horizons to evaluate adaptive strategies [2], and add diagnostic tasks to probe robustness, generalization under distributional shift, and few-shot performance [29, 23], enabling finer-grained analysis of algorithmic reliability in complex environments.

# 7 Conclusion

We introduced SVRPBench, an open benchmark for stochastic vehicle routing that integrates congestion dynamics, probabilistic delays, and heterogeneous time windows to mirror real-world logistics uncertainty. Across 500+ instances, classical and metaheuristic solvers remained robust in feasibility and runtime, while RL methods like POMO and AM underperformed under distributional shift, showing over 20% cost degradation. Multi-depot settings consistently improved cost and robustness, underscoring their practical value. By enabling large-scale, reproducible evaluation via Hugging Face and GitHub, SVRPBench establishes a community platform for developing adaptive, noise-aware routing algorithms that close the gap between synthetic optimization and real-world deployment.

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

# A  Open Infrastructure

To ensure reproducibility, extensibility, and accessibility, we release all components of the benchmark openly on GitHub and Hugging Face. This includes the dataset, instance generator, evaluation engine, and baseline implementations. Evaluation instances can be used out of the box, while the modular codebase allows users to integrate new solvers and adapt evaluation scripts.

A public leaderboard on huggingface[2] serves as the central hub for documentation, instance downloads, and leaderboard submissions. Submissions are validated automatically and ranked by total cost, feasibility, and runtime. All data and code are versioned, containerized (Docker-supported), and designed to support future extensions such as new routing scenarios or solver classes.

We welcome community contributions, including new solvers, datasets, and improvements to documentation or evaluation tools. By sharing the infrastructure broadly, we aim to foster collaboration and accelerate progress in realistic stochastic routing research.

## A.1  Reproducibility Requirements

To maintain transparency and enable fair comparison, submissions intended for leaderboard inclusion or academic publication must satisfy several criteria. Solvers must be evaluated on the official benchmark test set, with all hyperparameters, configuration details, and seed values fully documented. Additionally, we encourage open-source releases or detailed methodological descriptions to ensure algorithm reproducibility. Runtime should be measured using the official script or a clearly defined procedure, consistent across all experiments.

These guidelines help uphold reproducibility standards advocated in combinatorial optimization literature [9, 1] and promote meaningful scientific comparisons under controlled, yet realistic, conditions.

# B  Baseline Models

**Ant Colony Optimization (ACO).**  Routes are constructed by sampling next locations based on pheromone intensity and heuristic proximity. The pheromone matrix is updated as:

$$\tau_{ij} \leftarrow (1 - \rho)\tau_{ij} + \sum_{k=1}^{m} \Delta\tau_{ij}^{(k)}, \quad \Delta\tau_{ij}^{(k)} = \begin{cases} \frac{Q}{L^{(k)}}, & \text{if } (i,j) \in \text{tour}^{(k)} \\ 0, & \text{otherwise}, \end{cases} \tag{19}$$

where $\rho = 0.5$, $m = 50$ ants, $\alpha = 1$, and $\beta = 2$.

**Tabu Search.**  Candidate solutions are evaluated using a penalized cost function:

$$f(S) = \text{Cost}(S) + \lambda \cdot \text{Penalty}(S), \tag{20}$$

where $\lambda$ is adaptively tuned based on violation severity.

**Learning-Based Methods.**  The Attention Model is trained to minimize the expected cost:

$$\mathcal{L}(\theta) = \mathbb{E}_{X \sim \mathcal{D}} \left[ \mathbb{E}_{\pi_\theta(a|X)}[L(a|X)] \right]. \tag{21}$$

POMO uses multiple rollout agents initialized with distinct permutations. Its gradient signal is computed as:

$$\nabla_\theta J(\theta) = \frac{1}{M} \sum_{m=1}^{M} \sum_{t} \nabla_\theta \log \pi_\theta(a_t^m | s_t^m) \cdot (R^m - b), \tag{22}$$

where $M$ is the number of rollouts and $b$ is a learned baseline for variance reduction.

# C  Detailed Solver Performance Breakdowns

Tables 8,9,10,11,12,13 present a comprehensive performance breakdown of various solvers across multiple configurations for Capacitated VRP (CVRP) and Time Window VRP (TWVRP). Each solver,

---

[2]https://huggingface.co/spaces/ahmedheakl/SVRP-leaderboard

NN+2opt, Tabu Search, ACO, OR-Tools, and RL-based methods (Attention, POMO), is evaluated under different settings including depot configurations (single depot, multi depot, depots equal to cities), problem sizes (ranging from 10 to 1000 customers), and feasibility constraints. Metrics include total cost, CVR (constraint violation rate), feasibility, runtime, and time window violations. Traditional heuristic solvers (NN+2opt, Tabu, ACO) generally yield competitive costs with increasing runtimes as problem size grows, while OR-Tools offers consistent feasibility but with significantly higher runtimes. Reinforcement learning solvers (Attention, POMO) demonstrate exceptionally fast runtimes (in milliseconds), achieving full feasibility across all tested instances, although their cost can vary notably, especially for large-scale problems where some cost inflation is observed (e.g. POMO on 1000-node CVRP). These results highlight trade-offs between solution quality, computational efficiency, and scalability across solver paradigms.

Table 8: NN+2opt - Detailed Performance Breakdown.

| Configuration | Size | Cost | CVR | Feas | Runtime | TW Violations |
|---|---|---|---|---|---|---|
| single depot single vehicle sumDemands | 10 | 2290.7 | 0.0 | 1.000 | 0.0 | 0.00 |
| multi depot | 10 | 2371.8 | 0.0 | 1.000 | 2.0 | 0.00 |
| single depot single vehicle sumDemands | 20 | 3736.5 | 0.0 | 1.000 | 0.3 | 0.00 |
| multi depot | 20 | 3662.9 | 0.0 | 1.000 | 3.2 | 0.00 |
| single depot single vehicle sumDemands | 50 | 4840.4 | 0.0 | 1.000 | 10.5 | 0.00 |
| multi depot | 50 | 5626.1 | 0.0 | 1.000 | 14.1 | 0.00 |
| single depot single vehicle sumDemands | 100 | 6841.4 | 0.0 | 1.000 | 31.8 | 0.00 |
| multi depot | 100 | 7868.2 | 0.0 | 1.000 | 31.3 | 0.00 |
| single depot single vehicle sumDemands | 200 | 11268.2 | 0.0 | 1.000 | 125.2 | 0.00 |
| multi depot | 200 | 11479.2 | 0.0 | 1.000 | 135.5 | 0.00 |
| single depot single vehicle sumDemands | 500 | 16390.0 | 0.0 | 1.000 | 829.5 | 0.00 |
| multi depot | 500 | 17551.0 | 0.0 | 1.000 | 826.3 | 0.00 |
| single depot single vehicle sumDemands | 1000 | 25844.3 | 0.0 | 1.000 | 3545.9 | 0.00 |
| multi depot | 1000 | 25817.4 | 0.0 | 1.000 | 3493.3 | 0.00 |
| depots equal city | 10 | 4564.6 | 3.3 | 0.967 | 2.0 | 0.00 |
| single depot | 10 | 4359.0 | 3.3 | 0.967 | 2.0 | 0.00 |
| depots equal city | 20 | 8192.2 | 0.0 | 1.000 | 5.7 | 0.00 |
| single depot | 20 | 8347.0 | 0.0 | 1.000 | 5.0 | 0.00 |
| depots equal city | 50 | 13666.8 | 0.0 | 1.000 | 14.9 | 0.00 |
| single depot | 50 | 13882.4 | 0.7 | 0.993 | 11.7 | 0.00 |
| depots equal city | 100 | 38704.2 | 6.0 | 0.940 | 52.2 | 0.00 |
| single depot | 100 | 30389.4 | 1.0 | 0.990 | 37.8 | 0.00 |
| depots equal city | 200 | 89937.2 | 10.1 | 0.899 | 145.2 | 0.00 |
| single depot | 200 | 55400.9 | 1.0 | 0.990 | 167.8 | 0.00 |
| depots equal city | 500 | 175711.7 | 7.7 | 0.923 | 1318.5 | 0.00 |
| single depot | 500 | 118279.0 | 2.2 | 0.978 | 929.7 | 0.00 |
| depots equal city | 1000 | 244956.8 | 6.1 | 0.939 | 3865.2 | 0.00 |
| single depot | 1000 | 187829.7 | 2.7 | 0.973 | 3911.5 | 0.00 |

Table 9: Tabu Search - Detailed Performance Breakdown.

| Configuration | Size | Cost | CVR | Feas | Runtime | TW Violations |
|---|---|---|---|---|---|---|
| single depot single vehicle sumDemands | 10 | 2297.2 | 0.0 | 1.000 | 19.4 | 0.00 |
| multi depot | 10 | 2373.8 | 0.0 | 1.000 | 13.3 | 0.00 |
| single depot single vehicle sumDemands | 20 | 3776.7 | 0.0 | 1.000 | 47.6 | 0.00 |
| multi depot | 20 | 3656.4 | 0.0 | 1.000 | 33.8 | 0.00 |
| single depot single vehicle sumDemands | 50 | 4897.0 | 0.0 | 1.000 | 79.8 | 0.00 |
| multi depot | 50 | 5749.3 | 0.0 | 1.000 | 102.9 | 0.00 |
| single depot single vehicle sumDemands | 100 | 6981.9 | 0.0 | 1.000 | 170.0 | 0.00 |
| multi depot | 100 | 8058.6 | 0.0 | 1.000 | 169.2 | 0.00 |
| single depot single vehicle sumDemands | 200 | 11417.8 | 0.0 | 1.000 | 373.9 | 0.00 |
| multi depot | 200 | 11602.8 | 0.0 | 1.000 | 314.2 | 0.00 |
| single depot single vehicle sumDemands | 500 | 16554.8 | 0.0 | 1.000 | 1270.4 | 0.00 |
| multi depot | 500 | 17676.2 | 0.0 | 1.000 | 1445.1 | 0.00 |
| single depot single vehicle sumDemands | 1000 | 25995.4 | 0.0 | 1.000 | 4647.9 | 0.00 |
| multi depot | 1000 | 25879.7 | 0.0 | 1.000 | 4544.5 | 0.00 |
| depots equal city | 10 | 3966.1 | 3.3 | 0.667 | 185.9 | 0.00 |
| single depot | 10 | 4067.6 | 3.3 | 0.667 | 193.6 | 0.00 |
| depots equal city | 20 | 8156.1 | 0.0 | 1.000 | 479.8 | 0.00 |
| single depot | 20 | 7661.3 | 0.0 | 1.000 | 489.9 | 0.00 |
| depots equal city | 50 | 13918.7 | 0.0 | 1.000 | 719.3 | 0.00 |
| single depot | 50 | 14269.3 | 0.7 | 0.667 | 654.4 | 0.00 |
| depots equal city | 100 | 39031.2 | 6.0 | 0.000 | 2013.6 | 0.00 |
| single depot | 100 | 30820.4 | 1.0 | 0.333 | 1998.3 | 0.00 |
| depots equal city | 200 | 90028.5 | 10.1 | 0.000 | 2662.6 | 0.00 |
| single depot | 200 | 55596.2 | 1.0 | 0.000 | 3014.1 | 0.00 |
| depots equal city | 500 | 176001.3 | 8.1 | 0.000 | 13851.1 | 0.00 |
| single depot | 500 | 118726.0 | 2.2 | 0.000 | 11822.7 | 0.00 |
| depots equal city | 1000 | 244953.3 | 6.2 | 0.000 | 50402.1 | 0.00 |
| single depot | 1000 | 187945.6 | 2.7 | 0.000 | 42673.2 | 0.00 |

Table 10: ACO - Detailed Performance Breakdown.

| Configuration | Size | Cost | CVR | Feas | Runtime | TW Violations |
|---|---|---|---|---|---|---|
| single depot single vehicle sumDemands | 10 | 2183.6 | 0.0 | 1.000 | 14.3 | 0.00 |
| multi depot | 10 | 2325.4 | 0.0 | 1.000 | 11.9 | 0.00 |
| single depot single vehicle sumDemands | 20 | 3725.9 | 0.0 | 1.000 | 34.6 | 0.00 |
| multi depot | 20 | 3644.2 | 0.0 | 1.000 | 31.4 | 0.00 |
| single depot single vehicle sumDemands | 50 | 4840.5 | 0.0 | 1.000 | 165.2 | 0.00 |
| multi depot | 50 | 5626.2 | 0.0 | 1.000 | 179.5 | 0.00 |
| single depot single vehicle sumDemands | 100 | 6840.4 | 0.0 | 1.000 | 698.1 | 0.00 |
| multi depot | 100 | 7868.4 | 0.0 | 1.000 | 678.2 | 0.00 |
| single depot single vehicle sumDemands | 200 | 11264.3 | 0.0 | 1.000 | 2295.7 | 0.00 |
| multi depot | 200 | 11473.0 | 0.0 | 1.000 | 2380.3 | 0.00 |
| single depot single vehicle sumDemands | 500 | 16389.2 | 0.0 | 1.000 | 15573.5 | 0.00 |
| multi depot | 500 | 17551.6 | 0.0 | 1.000 | 16468.6 | 0.00 |
| single depot single vehicle sumDemands | 1000 | 25840.7 | 0.0 | 1.000 | 58364.4 | 0.00 |
| multi depot | 1000 | 25815.8 | 0.0 | 1.000 | 59341.2 | 0.00 |
| depots equal city | 10 | 3931.6 | 3.3 | 0.667 | 9.4 | 0.00 |
| single depot | 10 | 3819.2 | 3.3 | 0.667 | 9.6 | 0.00 |
| depots equal city | 20 | 7714.2 | 0.0 | 1.000 | 34.2 | 0.00 |
| single depot | 20 | 7749.4 | 0.0 | 1.000 | 34.1 | 0.00 |
| depots equal city | 50 | 13535.4 | 0.0 | 1.000 | 166.9 | 0.00 |
| single depot | 50 | 13872.4 | 0.7 | 0.667 | 143.6 | 0.00 |
| depots equal city | 100 | 37800.2 | 6.0 | 0.000 | 629.4 | 0.00 |
| single depot | 100 | 30389.5 | 1.0 | 0.333 | 679.0 | 0.00 |
| depots equal city | 200 | 89937.2 | 10.1 | 0.000 | 2556.8 | 0.00 |
| single depot | 200 | 55401.8 | 1.0 | 0.000 | 2327.0 | 0.00 |
| depots equal city | 500 | 175711.1 | 7.7 | 0.000 | 15299.3 | 0.00 |
| single depot | 500 | 118280.2 | 2.2 | 0.000 | 14781.5 | 0.00 |
| depots equal city | 1000 | 244999.0 | 6.1 | 0.000 | 70932.6 | 0.00 |
| single depot | 1000 | 187332.2 | 2.8 | 0.000 | 54846.8 | 0.00 |

Table 11: OR-Tools - Detailed Performance Breakdown.

| Configuration | Size | Cost | CVR | Feas | Runtime | TW Violations |
|---|---|---|---|---|---|---|
| single depot single vehicule sumDemands | 10 | 2049.2 | 0.0 | 1.000 | 1037.9 | 0.00 |
| multi depot | 10 | 2167.6 | 0.0 | 1.000 | 1003.3 | 0.00 |
| single depot single vehicule sumDemands | 20 | 3238.9 | 0.0 | 1.000 | 999.5 | 0.00 |
| multi depot | 20 | 3142.2 | 0.0 | 1.000 | 1002.6 | 0.00 |
| single depot single vehicule sumDemands | 50 | 3773.4 | 0.0 | 1.000 | 1015.9 | 0.00 |
| multi depot | 50 | 4714.2 | 0.0 | 1.000 | 1015.9 | 0.00 |
| single depot single vehicule sumDemands | 100 | 6283.5 | 0.0 | 1.000 | 1046.5 | 0.00 |
| multi depot | 100 | 6250.4 | 0.0 | 1.000 | 1048.8 | 0.00 |
| single depot single vehicule sumDemands | 200 | 9198.8 | 0.0 | 1.000 | 1174.7 | 0.00 |
| multi depot | 200 | 8956.2 | 0.0 | 1.000 | 1185.4 | 0.00 |
| single depot single vehicule sumDemands | 500 | 15677.5 | 0.0 | 1.000 | 2129.5 | 0.00 |
| multi depot | 500 | 15883.2 | 0.0 | 1.000 | 2085.2 | 0.00 |
| single depot single vehicule sumDemands | 1000 | 25844.3 | 0.0 | 1.000 | 8412.4 | 0.00 |
| multi depot | 1000 | 25816.3 | 0.0 | 1.000 | 9434.5 | 0.00 |
| depots equal city | 10 | 4564.7 | 3.3 | 0.967 | 12.6 | 0.00 |
| single depot | 10 | 4359.0 | 3.3 | 0.967 | 3.6 | 0.00 |
| depots equal city | 20 | 8192.3 | 0.0 | 1.000 | 8.2 | 0.00 |
| single depot | 20 | 8346.9 | 0.0 | 1.000 | 7.2 | 0.00 |
| depots equal city | 50 | 13666.7 | 0.0 | 1.000 | 30.3 | 0.00 |
| single depot | 50 | 13882.3 | 0.7 | 0.993 | 27.7 | 0.00 |
| depots equal city | 100 | 38704.1 | 6.0 | 0.940 | 108.6 | 0.00 |
| single depot | 100 | 30389.3 | 1.0 | 0.990 | 87.8 | 0.00 |
| depots equal city | 200 | 89937.5 | 10.1 | 0.899 | 345.3 | 0.00 |
| single depot | 200 | 55401.8 | 1.0 | 0.990 | 329.8 | 0.00 |
| depots equal city | 500 | 175711.4 | 7.7 | 0.923 | 2010.0 | 0.00 |
| single depot | 500 | 118279.4 | 2.2 | 0.978 | 2020.5 | 0.00 |
| depots equal city | 1000 | 244998.0 | 6.1 | 0.939 | 8273.1 | 0.00 |
| single depot | 1000 | 187830.1 | 2.7 | 0.973 | 8464.4 | 0.00 |

Table 12: RL Algorithms – Detailed Performance on CVRP (runtimes in ms).

| Solver | Configuration | Size | Cost | CVR | Feas | Runtime (ms) | TW Violations |
|---|---|---|---|---|---|---|---|
| Attention | single depot single vehicule sumDemands | 10 | 2364.12 | 0.00 | 1.000 | 0.365 | 0.00 |
| POMO | single depot single vehicule sumDemands | 10 | 2312.68 | 0.00 | 1.000 | 0.282 | 0.00 |
| Attention | single depot single vehicule sumDemands | 20 | 3222.68 | 0.00 | 1.000 | 0.269 | 0.00 |
| POMO | single depot single vehicule sumDemands | 20 | 3341.56 | 0.00 | 1.000 | 0.279 | 0.00 |
| Attention | single depot single vehicule sumDemands | 50 | 5803.63 | 0.00 | 1.000 | 0.304 | 0.00 |
| POMO | single depot single vehicule sumDemands | 50 | 5920.19 | 0.00 | 1.000 | 0.287 | 0.00 |
| Attention | single depot single vehicule sumDemands | 100 | 8553.26 | 0.00 | 1.000 | 0.319 | 0.00 |
| POMO | single depot single vehicule sumDemands | 100 | 16983.50 | 0.00 | 1.000 | 0.319 | 0.00 |
| Attention | single depot single vehicule sumDemands | 200 | 13228.84 | 0.00 | 1.000 | 0.353 | 0.00 |
| POMO | single depot single vehicule sumDemands | 200 | 12726.96 | 0.00 | 1.000 | 0.360 | 0.00 |
| Attention | single depot single vehicule sumDemands | 500 | 22496.94 | 0.00 | 1.000 | 0.463 | 0.00 |
| POMO | single depot single vehicule sumDemands | 500 | 88789.44 | 0.00 | 1.000 | 0.506 | 0.00 |
| Attention | single depot single vehicule sumDemands | 1000 | 37430.47 | 0.00 | 1.000 | 0.649 | 0.00 |
| POMO | single depot single vehicule sumDemands | 1000 | 184656.10 | 0.00 | 1.000 | 0.689 | 0.00 |

Table 13: RL Algorithms – Detailed Performance on TWVRP (runtimes in ms).

| Solver | Configuration | Size | Cost | CVR | Feas | Runtime (ms) | TW Violations |
|---|---|---|---|---|---|---|---|
| Attention | single depot | 10 | 3 940.38 | 0.00 | 1.000 | 0.916 | 0.00 |
| POMO | single depot | 10 | 3 854.6 | 0.00 | 1.000 | 0.707 | 0.00 |
| Attention | single depot | 20 | 6 504.73 | 0.00 | 1.000 | 1.780 | 0.00 |
| POMO | single depot | 20 | 6 744.7 | 0.00 | 1.000 | 1.841 | 0.00 |
| Attention | single depot | 50 | 29 132.94 | 0.00 | 1.000 | 0.731 | 0.00 |
| POMO | single depot | 50 | 29 718.0 | 0.00 | 1.000 | 0.689 | 0.00 |
| Attention | single depot | 100 | 57 778.84 | 0.00 | 1.000 | 0.864 | 0.00 |
| POMO | single depot | 100 | 114 726.7 | 0.00 | 1.000 | 0.864 | 0.00 |
| Attention | single depot | 200 | 113 742.27 | 0.00 | 1.000 | 0.868 | 0.00 |
| POMO | single depot | 200 | 109 427.1 | 0.00 | 1.000 | 0.886 | 0.00 |
| Attention | single depot | 500 | 271 201.60 | 0.00 | 1.000 | 1.412 | 0.00 |
| POMO | single depot | 500 | 438 502.6 | 0.00 | 1.000 | 1.412 | 0.00 |
| Attention | single depot | 1000 | 531 470.88 | 0.00 | 1.000 | 1.638 | 0.00 |
| POMO | single depot | 1000 | 611 307.8 | 0.00 | 1.000 | 1.672 | 0.00 |

## C.1 Qualitative Results

As shown in figures 5, 6, 7, 8, 9, 10, 11, 12, 13, and 14, we qualitatively observe that for CVRP instances with a small number of customers, both Attention and POMO models, as well as classical methods (ACO, NN2OPT, and OR-Tools), generate highly structured and near-optimal routes. As the number of customers increases, route complexity grows, making it harder for models to preserve efficiency and structure. For TWVRP, the models' priority shifts toward satisfying delivery time windows, often at the expense of distance optimization. This results in routes that appear less spatially coherent but better aligned with temporal constraints.

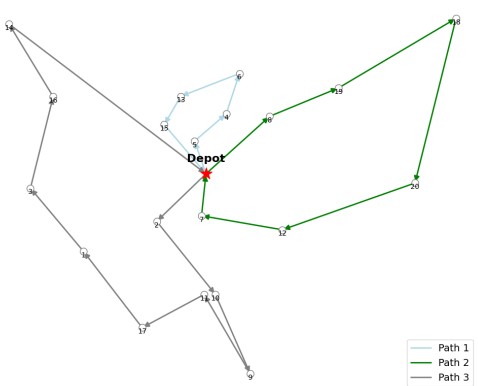

Figure 5: CVRP 20 customers – Attention Model

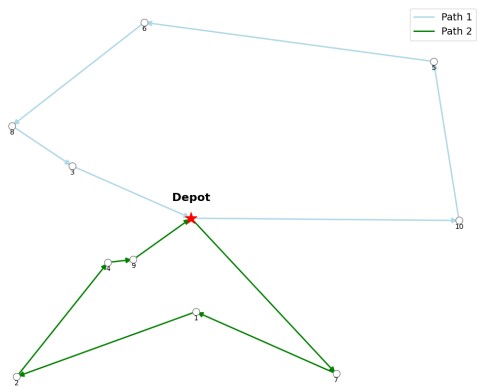

Figure 6: CVRP 10 customers – POMO

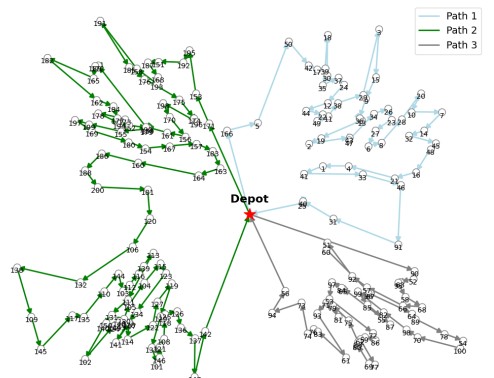

Figure 7: CVRP 200 customers – Attention Model

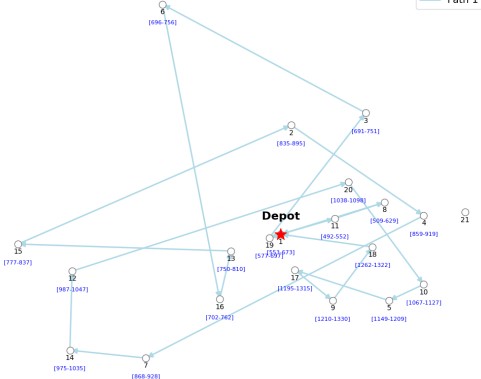

Figure 8: TWVRP 20 customers – Attention Model

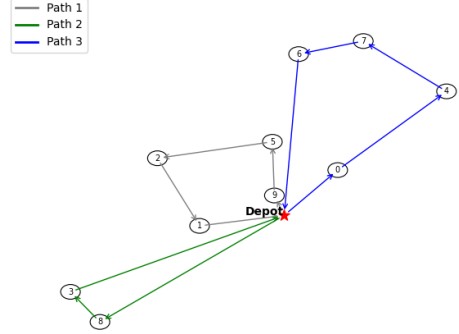

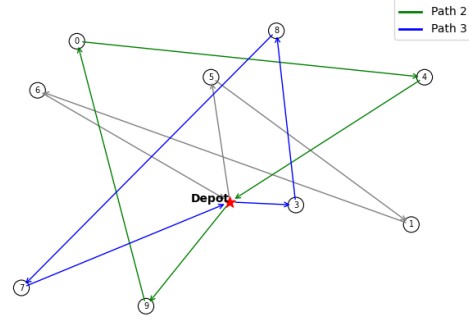

Figure 9: CVRP 10 customers – ACO

Figure 10: TWVRP 10 customers – ACO

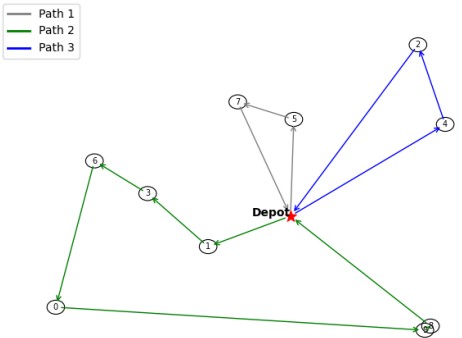

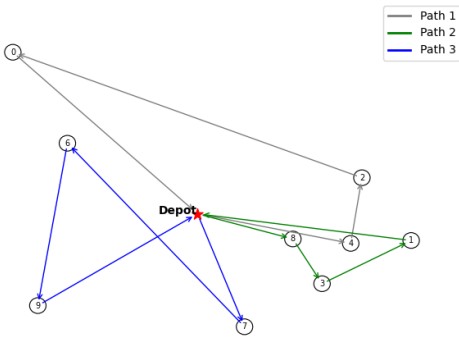

Figure 11: CVRP 10 customers – NN2OPT

Figure 12: TWVRP 10 customers – NN2OPT

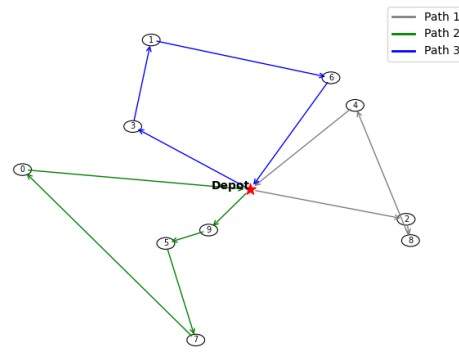

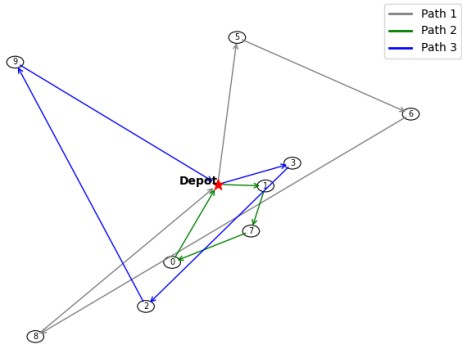

Figure 13: CVRP 10 customers – OR-Tools

Figure 14: TWVRP 10 customers – OR-Tools

# D  Reinforcement Learning

## D.1  Problem Formulation

We model both the Capacitated Vehicle Routing Problem (CVRP) and Vehicle Routing Problem with Time Windows (VRPTW) as a Markov Decision Process (MDP) $\mathcal{M} = (\mathcal{S}, \mathcal{A}, P, r, \gamma)$, where each state $s_t \in \mathcal{S}$ encodes the vehicle's current position, remaining capacity, visited set (and only for VRPTW the current time and per-customer time windows $[e_i, \ell_i]$). Actions $a_t \in \mathcal{A}(s_t)$ select the next customer, and transitions $P(s_{t+1} \mid s_t, a_t)$ deterministically update the tour while, in VRPTW, adding stochastic delays.

The reward is $r(s_t, a_t) = -d_{i,j} - \tau\, [t_{\text{arrive}} > \ell_i]$ when visiting customer $j$, with $d_{i,j}$ the Euclidean distance and $\tau$ a large penalty for time-window violations, and zero upon return to the depot. We follow a constructive, autoregressive decoding: at each step we append one customer until all are visited.

## D.2  Policy

We adopt the encoder–decoder with multi-head attention of Kool [22]. Given embedded node features $\mathbf{x}_i \in \mathbb{R}^d$, each of the $L$ encoder layers applies multi-head self-attention. At step $t$, with context embedding $\mathbf{h}_t$, we score each remaining node $j$ by $u_{t,j} = \mathbf{v}^\top \tanh(W_1 \mathbf{h}_t + W_2 \mathbf{x}_j)$ and define $\pi_\theta(a_t = j \mid s_t) = \exp(u_{t,j}) / \sum_{k \notin \mathcal{V}_t} \exp(u_{t,k})$.

We optimize the policy by maximizing the expected return $J(\theta) = \mathbb{E}_{\tau \sim \pi_\theta}[R(\tau)]$ using two constructive, autoregressive policy-gradient methods. A constructive policy builds a complete solution by sequentially selecting one customer at a time until the tour is finished, while an autoregressive policy conditions each action on the history of previous choices, enabling the network to capture dependencies across steps.

We first apply REINFORCE [40], which updates parameters via $\nabla_\theta J(\theta) = \mathbb{E}\big[\sum_t \nabla_\theta \log \pi_\theta(a_t \mid s_t)\,(R(\tau) - b(s_t))\big]$, where $b(s_t)$ is a rollout baseline obtained by greedy decoding; then POMO [23] samples $K$ different start nodes per instance, computes returns $R_k$ and a shared baseline $\bar{R} = \frac{1}{K} \sum_k R_k$, and applies $\nabla_\theta J(\theta) = \frac{1}{K} \sum_{k=1}^{K} \nabla_\theta \log \pi_\theta(\tau_k)\,(R_k - \bar{R})$. REINFORCE offers simplicity and unbiased gradients, while POMO's shared baseline exploits VRP permutation symmetry for variance reduction; together they provide a strong comparison between a classical Monte Carlo approach and a state-of-the-art, variance-reduced VRP-specific algorithm.

## D.3  Training Details

All models were implemented in the RL4CO framework and trained end-to-end with Adam at a learning rate of $10^{-4}$. For CVRP with REINFORCE we used a batch size of 512 and generated 100 000 synthetic instances on the fly; for VRPTW with POMO we used batch size 64 and 1 000 000 instances. Validation employed greedy decoding under nominal travel-time conditions. VRPTW environments included log-normal delays calibrated to traffic data, Gaussian time-of-day kernels, and Poisson accident events, with infeasible actions heavily penalized to enforce time windows.

## D.4  Evaluation on SVRPBench

After training, we converted each of the 500+ SVRPBench instances into the RL4CO environment format and ran the trained policies in greedy mode, selecting at each step $a_t = \arg\max_j \pi_\theta(a_t = j \mid s_t)$. To assess robustness, we then simulated each resulting tour under multiple sampled delay realizations and reported average tour length and feasibility rates. Despite domain shift, attention-based RL policies maintained high feasibility and near-optimal costs across all problem sizes.

