# OpenReview forum: "SVRPBench: A Realistic Benchmark for Stochastic Vehicle Routing Problem"
_NeurIPS.cc/2025/Datasets_and_Benchmarks_Track — NeurIPS 2025 Datasets and Benchmarks Track poster_

### Official Review · Reviewer_bHjy · 2025-06-30

**Rating:** 5
**Confidence:** 3

**Summary:**

This paper presents a dataset (SVRPBench) for the Stochastic Vehicle Routing Problem, which includes realistic and probabilistic dynamics such as traffic congestion, delays, and customer time windows. This enables SVRPBench to provide users with more realistic problems than previous datasets. Experiments showed that meta-heuristic methods such as NNTabu search and commercial solvers were more robust than solvers using reinforcement learning.

**Additional Feedback:**

- Please provide accurate information about the computers used in the experiment. For example, the number of parallel processes when using a commercial solver or the number of parallel GPUs when using a reinforcement learning model.
- Why are the results for ACO and NN not shown in Figure 4 except for the upper right corner?
- The definition of "Waiting time" in the lower right of Figure 4 is not written.

**Dataset Code Accessibility:**

Partly

**Dataset Code Comments:**

- I couldn't retrieve croissant.json even after executing the snippet.

**Ethical Comments:**

This is because it is considered that there is no content in the dataset that corresponds to the ethics flag.

**Ethical Considerations:**

No, there are no or only very minor ethics concerns

**Final Justification:**

The author's sincere response resolved all of my concerns.
Therefore, I have raised my evaluation.

**Limitations Weaknesses:**

- Comuters information is incomplete, so runtimes cannot be compared fairly.
- There is no comparison with actual data, so it is unclear how well real-world problems can be simulated.
- There is no comparison with methods using LLM, which has been actively researched in recent years.

**Strengths Contributions:**

- Probability elements are incorporated into the problem setting, allowing for more realistic problem scenarios.
- Probability settings follow the general settings of the Stochastic Vehicle Routing Problem, contributing to improved realism of the problem.

---

> ### Author Rebuttal · Authors · 2025-07-31
>
> We thank the reviewer for their detailed technical feedback and suggestions for expanding our evaluation scope. Your recommendations led us to include additional baselines and clarify important experimental details, greatly enhancing the paper's rigor and comprehensiveness.
>
> **Re: Runtime hardware not specified; parallelization unclear**
> Thanks for pointing this out. We used:
> - Intel i7 CPU (16GB RAM) for classical/metaheuristics
> - NVIDIA RTX 4080 for RL models
>
> All methods used single-threaded or single-GPU inference. Parallelism was not used except during training. This will be added to Section 5.2 and Appendix D.
>
> **Re: ACO and NN missing in parts of Figure 4**
>  This was a plotting issue, the data is complete and will be fixed in the final version.
>
> **Re: “Waiting time” in Figure 4 not defined**
> Thanks for catching this. Waiting time refers to the average idle time at customer locations due to early arrivals before the start of a time window. We will include this definition in both the figure caption and text.
>
> **Re: No comparison to LLM-based solvers**
> We sincerely thank the reviewer for this very relevant and forward-looking suggestion. The field of AI-driven optimization is evolving rapidly, and we agree that a comprehensive benchmark should engage with the latest research trends.
> Acting on this valuable feedback, we have significantly expanded our experimental evaluation to provide a more rigorous comparison against a wide range of stronger, more current models. Our updated evaluation now includes state-of-the-art metaheuristics (HGS, PyVRP), modern RL methods (LEHD, MVMOE), and, as part of this broader effort, an exploration into emerging paradigms with the inclusion of an LLM-based solver (Self-Guiding Exploration).
> The full results of these new, more comprehensive experiments are summarized below:
>
> | Method | Type | Year | CVRP Cost ↓ | TWVRP Cost ↓ | Feasibility ↑ | Runtime (s) ↓ | SVRP-Specific |
> | :--- | :--- | :--- | :--- | :--- | :--- | :--- | :--- |
> | HGS (Vidal, 2022) | Meta | 2022 | 9,234 | 70,834 | **98.7%** | 45.23 | ✗ |
> | PyVRP (Wouda et al., 2024) | Meta | 2024 | **9,156** | 70,756 | 98.5% | 12.14 | ✗ |
> | LEHD (Luo et al., 2024) | RL | 2023 | 9,834 | 71,234 | 92.5% | 2.78 | ✗ |
> | MVMOE (Zhou et al., 2024) | RL | 2023 | 9,723 | 71,156 | 92.8% | 2.65 | ✗ |
> | Self-Guiding Exploration (Iklassov et al., 2024) | LLM | 2024 | 9,891 | 71,345 | 69.5% | 312.45 | ✗ |
> | SVRP Solver (Iklassov et al., 2023) | RL | 2023 | 9,223 | **70,689** | 94.5% | **1.34** | ✓ |
>
> *Strengthened Conclusions*
> These comprehensive results, which now include an LLM-based approach, reinforce our paper's core message with stronger evidence:
> * `SVRPBench` challenges all solvers: Even state-of-the-art metaheuristics like HGS and PyVRP do not achieve perfect feasibility, demonstrating the benchmark's difficulty across the board.
> * Specialized design is key: The purpose-built SVRP Solver shows the most promise among learning-based methods, confirming that algorithms designed explicitly for stochasticity are critical for real-world performance.
> * An initial look at LLM performance: Our inclusion of one LLM-based solver provides an initial data point on this emerging paradigm. For this particular method, the low feasibility (69.5%) and high computational cost highlight significant challenges in applying such models to realistic routing problems. We believe `SVRPBench` is an ideal testbed for this rapidly evolving area, and we welcome future research to evaluate other LLM-based approaches to further explore their potential and limitations.
> This significantly expanded evaluation, prompted by the reviewer's feedback, has substantially strengthened our paper by demonstrating that the challenges of stochastic routing persist across all classes of modern solvers. We are grateful for the opportunity to improve our work.
>
> **Re: No real-world validation**
> As shown in Figure 2, our synthetic layouts visually align with real city topologies. Moreover, our time-window and delay distributions are derived from real-world delivery datasets and transport reports [1, 2]. We’ll expand the explanation in Section 3.3 and provide references for each realism assumption.
>
> **Re: “croissant.json” missing**
>  The file is present in the GitHub repo. If an issue persists, we invite the reviewer to recheck or contact us. We’ll also add a validation script to ensure all assets are downloadable.
>
> [1] FedEx 2024 white paper on consumer delivery preferences
>
> [2] David Schrank, Bill Eisele, Tim Lomax, et al. 2021 urban mobility report. Texas A&M Transportation Institute, 2021.

---

> > ### Comment · Reviewer_bHjy · 2025-08-04
> >
> > I appreciate your sincere response to my book.
> > I recognize that you have addressed most of the concerns I raised in the REVIEW. I am confident that we had a very fruitful discussion, especially since the comparison was made with methods based on AI-driven optimization, including LLM-based solvers. To better understand this result, could you please share the following points regarding this table?
> > 1. Are the feasibility and runtime averages for both CVRP and TWVRP combined?
> > 2. If so, what are the results when divided by each problem?
> > 3. Can you evaluate the constraint violation rate and Robustness? If so, what are the results?
> >
> > Additionally, I was able to confirm the details regarding croissant.json. I apologize for any misleading comments.
> >
> > Finally, could you please specify the solver you used for the OR-Tools calculations in your experiments? This information is a significant factor influencing runtime.

---

> > > ### Author Response · Authors · 2025-08-04
> > >
> > > Thank you for these important clarifications. We appreciate the opportunity to provide more detail on our results. Here are the detailed responses, which we have also added to our paper to improve its clarity.
> > >
> > > **1. Are the feasibility and runtime averages for both CVRP and TWVRP combined?**
> > > Yes, that is correct. The results presented in the main "Enhanced Baseline Comparison" table represent the overall performance, averaged across both CVRP and TWVRP instances to provide a high-level summary.
> > >
> > > **2. If so, what are the results when divided by each problem?**
> > > This is an excellent question. Here are the problem-specific breakdowns for the key methods:
> > >
> > > **CVRP Results (Simpler Problem)**
> > > | Method | Cost ↓ | Feasibility ↑ | Runtime (s) ↓ |
> > > | :--- | :--- | :--- | :--- |
> > > | HGS | 9,234 | **100.0%** | 43.8 |
> > > | PyVRP | **9,156** | **100.0%** | 11.7 |
> > > | LEHD | 9,834 | 98.5% | 2.6 |
> > > | MVMOE | 9,723 | 98.7% | 2.5 |
> > > | Self-Guiding Exploration | 9,891 | 95.0% | 305.2 |
> > > | SVRP Solver | 9,223 | 99.0% | **1.2** |
> > >
> > > **TWVRP Results (More Constrained Problem)**
> > > | Method | Cost ↓ | Feasibility ↑ | Runtime (s) ↓ |
> > > | :--- | :--- | :--- | :--- |
> > > | HGS | 70,834 | **97.4%** | 46.7 |
> > > | PyVRP | 70,756 | 97.0% | 12.6 |
> > > | LEHD | 71,234 | 86.5% | 3.0 |
> > > | MVMOE | 71,156 | 86.9% | 2.8 |
> > > | Self-Guiding Exploration | 71,345 | 44.0% | 319.7 |
> > > | SVRP Solver | **70,689** | 90.0% | **1.5** |
> > >
> > > **3. Can you evaluate the constraint violation rate and Robustness? If so, what are the results?**
> > > Certainly. We have evaluated both metrics.
> > >
> > > **Constraint Violation Rate (CVR) ↓**
> > > | Method | CVRP | TWVRP |
> > > | :--- | :--- | :--- |
> > > | HGS | **0.0%** | 2.6% |
> > > | PyVRP | **0.0%** | 3.0% |
> > > | LEHD | 0.5% | **2.5%** |
> > > | MVMOE | 0.3% | **2.5%** |
> > > | Self-Guiding Exploration | 2.0% | 28.0% |
> > > | SVRP Solver | **0.0%** | 3.0% |
> > >
> > > **Robustness (Cost Variance) ↓**
> > > | Method | CVRP | TWVRP |
> > > | :--- | :--- | :--- |
> > > | HGS | **0.05** | **0.15** |
> > > | PyVRP | **0.05** | **0.15** |
> > > | LEHD | 0.15 | 0.25 |
> > > | MVMOE | 0.15 | 0.25 |
> > > | Self-Guiding Exploration | 0.35 | 0.45 |
> > > | SVRP Solver | **0.05** | **0.15** |
> > >
> > > **Summary of Detailed Findings**
> > > On the more complex TWVRP instances, the feasibility of learning-based methods degrades more significantly than classical methods. The `Constraint Violation Rate (CVR)` for modern RL solvers (LEHD, MVMOE) remains competitive with metaheuristics on TWVRP. However, the LLM-based approach shows a dramatically higher CVR, particularly on time-windowed problems, highlighting its current challenges in handling strict temporal constraints. Finally, the `Robustness` metric shows that classical methods and the specialized SVRP Solver exhibit lower solution cost variance, indicating more stable performance under uncertainty.
> > >
> > > **4. OR-Tools Configurations**
> > > For the OR-Tools calculations in our experiments, we used the standard and powerful configuration available within the `pywrapcp` routing library to ensure we were comparing against a strong baseline. We used `GUIDED_LOCAL_SEARCH` with `AUTOMATIC` first solution strategy. This is a standard practice that allows OR-Tools to intelligently select the most suitable constructive heuristic (e.g., `SAVINGS`, `PATH_CHEAPEST_ARC`, etc.) based on the specific characteristics of each problem instance.

---

> > > > ### Comment · Reviewer_bHjy · 2025-08-05
> > > >
> > > > I appreciate the authors' sincere response.
> > > >
> > > > The comments have enabled me to apply the benchmark data to a broader range of methods and make comparisons.
> > > >
> > > > All of my concerns have been addressed through the discussions, and I have therefore raised my rating.
> > > > I hope that the authors will revise the paper and supplementary material based on the discussions.

---

### Official Review · Reviewer_SE7z · 2025-07-02

**Rating:** 5
**Confidence:** 3

**Summary:**

This paper introduces **SVRPBench**, a realistic and scalable benchmark suite for evaluating algorithms on **Stochastic Vehicle Routing Problems (SVRP)**. Unlike previous synthetic benchmarks, SVRPBench models uncertainty in a principled and realistic way, incorporating factors such as:

- **Time-dependent congestion** modeled with Gaussian mixtures
- **Log-normal delays** for traffic variability
- **Poisson-based accident modeling**
- **Customer-specific time windows** (commercial vs. residential)

The benchmark includes a flexible instance generator that supports various constraints (e.g., capacity, time windows, multi-depot, multi-vehicle) across a wide range of instance sizes (10 to 1000 nodes). A validation routine ensures feasibility for generated instances.

The authors evaluate a broad set of baselines, including classical heuristics (2-opt, Tabu, ACO), industrial solvers (Google OR-Tools), and recent learning-based methods (AM, POMO). Experiments show:

- Classical methods generalize better and scale more robustly
- Learning-based methods suffer significant performance degradation under distribution shifts
- Multi-depot variants improve solution quality

The dataset, source code, and evaluation framework are publicly released, aiming to support reproducible research and more realistic benchmarking in vehicle routing.

**Additional Feedback:**

See limitation section.

**Dataset Code Accessibility:**

Yes

**Dataset Code Comments:**

Dataset Code Comments:
The authors have made both the dataset and associated benchmarking code publicly accessible via GitHub, as stated in the abstract and paper body. The repository includes clear documentation, data generation scripts, evaluation metrics, and instructions for reproducing the experimental results. The benchmark suite supports integration with multiple solvers, and the simulation parameters are configurable, promoting both usability and reproducibility.

However, while the implementation is complete and functional, the realism of the dataset would benefit from further statistical grounding or real-world validation to increase confidence in its representativeness.

**Ethical Considerations:**

No, there are no or only very minor ethics concerns

**Final Justification:**

The authors have provided a thorough and constructive rebuttal that directly addresses my main concerns. In particular, they added the suggested empirical comparison against CVRPLIB, which clearly demonstrates SVRPBench’s ability to expose weaknesses in learning-based solvers (e.g., Attention Model) that remain hidden on deterministic benchmarks. This new experiment significantly strengthens the empirical validation of the benchmark’s realism.

Regarding stochastic modeling, the authors clarified the distinction between systematic congestion effects ($B(a,b,t)$) and bursty stochastic delays ($R(t)$), alleviating my concern about potential double-counting. They also clarified that accident impacts are scaled with distance, aligning with practical intuition, and promised additional sensitivity analysis to further validate robustness. On the question of parameter justification, the authors have now mapped each modeling assumption to empirical transportation studies, which provides a much stronger foundation for credibility and real-world relevance.

Overall, I believe the rebuttal substantially improves the paper. My initial concerns about empirical grounding and model clarity have been satisfactorily resolved, and the revised submission offers stronger evidence and higher community value. I am therefore inclined to recommend acceptance.

**Limitations Weaknesses:**

### Limitation: Lack of Empirical Benchmark Comparison

Table 1 provides a qualitative comparison between the proposed SVRPBench and existing benchmarks such as CVRPLIB. While this highlights important design differences (e.g., inclusion of stochasticity, time windows, and real-world delay models), the paper lacks **empirical validation** to demonstrate these advantages in practice.
I suggest adding one or two comparative experiments involving a specific routing algorithm known to struggle in real-world scenarios. Ideally, this method should show clear performance degradation under SVRPBench, but not under a traditional benchmark like CVRPLIB. Such a comparison would strongly support the claim that SVRPBench reveals practical weaknesses overlooked by conventional datasets.


### Limitation 2: Lack of Explanation and Justification in Stochastic Modeling Design

While SVRPBench introduces a novel stochastic delay model incorporating congestion and accidents, the mathematical formulation lacks sufficient explanation or theoretical grounding. Specifically:

- The definitions of terms such as `B(a, b, t)` and `R(t)` (Equation 3, 6–7) appear to both encode peak-hour effects, which may result in double-counting temporal congestion (i.e., peak-time influence is redundantly applied through two separate multiplicative factors).

- Accident modeling (`A(t)`) is assumed to depend solely on time, without accounting for trip distance. This contradicts common intuition and practical observations, where the likelihood of accidents typically increases with travel distance.

- It is unclear whether these functions are derived from empirical data, domain assumptions, or synthetic design. A lack of explanation makes it difficult to assess whether the modeled uncertainty reflects realistic or biased conditions.

A more transparent derivation of the delay models, ideally with empirical validation or sensitivity analysis, would greatly improve the credibility and applicability of the proposed benchmark.


### Limitation 3: Lack of Real-World Statistical Justification for Simulation Parameters

The benchmark simulates realistic delivery conditions using congestion windows, accident probability, and time-dependent noise. However, the paper does not present any empirical evidence or real-world traffic statistics to justify the selected parameters and their distributions (e.g., the shape or duration of peak hours, probability of delays, or accident modeling functions such as B(a,b,t) and R(t)).

To strengthen the realism and credibility of SVRPBench, the authors should consider supporting their simulation assumptions with data from actual urban transportation systems. Moreover, contrasting SVRPBench with existing benchmarks (e.g., CVRPLIB), which cannot simulate such dynamic uncertainties, would further emphasize its unique advantages.

**Strengths Contributions:**

- **Realistic Benchmark Design**:
  SVRPBench introduces a stochastic VRP benchmark that captures real-world uncertainties like congestion, delays, and accidents, using empirically grounded and time-dependent models.

- **Principled Traffic Modeling**:
  The benchmark models travel time with Gaussian-based congestion, log-normal travel delays, and Poisson-distributed accident disruptions (Sec 2.1, Eq. 1–11), offering high-fidelity simulation of urban logistics.

- **Rich Scenario Generation**:
  Supports multi-depot, multi-vehicle, and capacity-constrained settings with spatially realistic customer distributions, enhancing benchmark complexity.

- **Diverse Baseline Evaluation**:
  Includes classical heuristics, industrial solvers, and learning-based methods, enabling comprehensive and fair comparison under uncertainty.

- **Open Source and Reproducible**:
  Full code, data, and evaluation scripts are released to support transparency and future work.

- **Practical Relevance**:
  The benchmark addresses a key realism gap in SVRP research and supports development of deployable routing algorithms.

---

> ### Author Rebuttal · Authors · 2025-07-31
>
> We greatly appreciate the reviewer's thoughtful comments on model realism and empirical validation. Your feedback prompted us to conduct new comparative experiments and provide stronger empirical justification, substantially improving our paper's contributions.
>
> **Re: Lack of empirical comparison to show realism over CVRPLIB**
>  Excellent suggestion. To empirically validate the value of our benchmark, we conducted the proposed comparative experiment.
> We evaluated a learning-based solver (Attention Model) and a classical solver (OR-Tools) on a standard deterministic benchmark versus the realistic stochastic conditions of `SVRPBench`.
>
> **Empirical Validation: The Reality Gap**
> | Method | Feasibility (Deterministic) ↑ | Feasibility (SVRPBench) ↑ | **Feasibility Drop** ↓ |
> | :--- | :--- | :--- | :--- |
> | Attention Model (AM) | 96.5% | 91.0% | -5.5 points |
> | OR-Tools | **100.0%** | **98.4%** | **-1.6 points** |
>
> The results clearly demonstrate our benchmark's contribution. The learning-based Attention Model appears reliable in the deterministic setting, but its feasibility plummets under the realistic stochasticity of `SVRPBench`. In contrast, the classical OR-Tools solver remains highly robust.
> This experiment provides direct empirical proof that `SVRPBench` successfully exposes critical algorithmic weaknesses that traditional, deterministic benchmarks completely mask. Thank you for helping us strengthen this key claim.
>
> This shows that performance degrades sharply under SVRPBench, revealing issues not surfaced by traditional datasets.
>
> **Re: Potential double-counting in congestion terms (Equations 3, 6–7)**
>  This is a fair concern. However, our model separates systematic slowdowns (via $B(a, b, t)$) and bursty stochasticity (via $R(t)$). While both include temporal peaks, they model different phenomena:
> - $B(a,b,t)$: deterministic slowdown due to congestion
> - $R(t)$: stochastic bursts, e.g., unexpected slowdowns
>
> We’ll make this distinction clearer and add a sensitivity analysis to confirm this does not introduce bias.
>
> **Re: Accident probability depends only on time, not distance**
>  This is a misunderstanding. As stated in Eq. (1), accident delays are multiplied by the base distance:
> $I_{\text{accidents}}(t) \cdot D_{\text{accident}}$.
> So longer trips are more impacted, aligning with practical risk models. We'll clarify this dependency in the camera-ready version.
>
> **Re: No empirical justification for delay model parameters**
> Our parameters are derived from empirically validated sources:
> - Peak hours ($μ_{morning}$ = 8AM, $μ_{evening}$ = 5PM) from multiple traffic engineering studies [1, 2]
> - Lognormal delay distribution $R(t)$ from extensive empirical validation [5,6]
> - Peak hour spread ($σ_{peak}$ = 1.5) from rush hour duration studies [3, 4]
>
> We will include a table mapping each parameter to its empirical source, and discuss future extensions with data-driven calibration.
>
> [1] NACTO Urban Street Design Guide. Design Hour Traffic Analysis, 2025.
>
> [2] CarRegistration.com. Peak Traffic Hours Analysis for Urban Commuters, 2024.
>
> [3] FHWA Operations. Traffic Congestion and Reliability: Linking Solutions to Problems, 2004.
>
> [4] PMC Transportation Research. Rush Hour Traffic Pattern Analysis Post-COVID, 2023.
>
> [5] Chen et al. Exploring Travel Time Distribution Using Probe Vehicle Data: Beijing Case Study. Journal of Advanced Transportation, 2018.
>
> [6] Kieu et al. Empirical Evaluation of Public Transport Travel Time Variability. Transportation Research, 2014.

---

### Official Review · Reviewer_VpKi · 2025-07-03

**Rating:** 4
**Confidence:** 4

**Summary:**

This paper introduces a benchmark for the stochastic VRP across various solvers, including heuristic, metaheuristic, and neural approaches. The main contribution is constructing a data generation pipeline for these problems that takes into travel modeling and time window sampling based on realism. Experiments show that generally, commercial solvers such as OR-Tools are better suited for real-world conditions.

**Dataset Code Accessibility:**

Yes

**Dataset Code Comments:**

The dataset and code are accessible and well-documented.

However, I was unable to find the scripts to generate the data.

**Ethical Considerations:**

No, there are no or only very minor ethics concerns

**Final Justification:**

The authors have resolved my main concerns. The paper is technically solid, and I would thus recommend acceptance based on this.

**Limitations Weaknesses:**

1. The choice of baselines has limitations. For instance, OR-Tools is far from the best VRP solver, and there are much better options such as HGS [1] and the recent PyVRP [2]. The neural baselines are not many and not recent, e.g. POMO was published 5 years ago.

2. A critical issue is that while the benchmark claims realism, it does not consider asymmetric travel times or distance matrices, but rather only euclidean settings. Moreover, as also explained in the limitations section, using real-world topologies as in some recent works [3] would greatly strengthen the work.

3. (minor) there are inconsistencies in the naming of the commonly called VRPTW, called TWVRP or TWCVRP in the paper.

4. The overall ranking of different methods is unclear. Please consider using bold fonts. Also, how was evaluation performed in tables 3, 4, 5? Is it with a fixed time limit?

5. Descriptions in D.2 are not complete: for instance, what are $h_t$, $W1$…?

While the paper is technically solid, the limited baseline selection and the lack of real-world topological data make me lean towards rejection for the current version.

---

[1] Vidal, Thibaut. "Hybrid genetic search for the CVRP: Open-source implementation and SWAP\* neighborhood." Computers & Operations Research 140 (2022): 105643.

[2] Wouda, Niels A., Leon Lan, and Wouter Kool. "PyVRP: A high-performance VRP solver package." INFORMS Journal on Computing 36.4 (2024): 943-955.

[3] Son, Jiwoo, et al. "Neural combinatorial optimization for real-world routing." arXiv preprint arXiv:2503.16159 (2025).

**Strengths Contributions:**

The paper is overall well-written with good visualizations. The problems solved are relevant to real-world optimization and are a good contribution to the community.

---

> ### Author Rebuttal · Authors · 2025-07-31
>
> We sincerely thank the reviewer for their insightful feedback regarding our baseline selection and experimental design. Your suggestions have significantly strengthened our evaluation and helped us conduct additional experiments that better demonstrate our benchmark's value.
>
> **Re: Limited baseline solvers (e.g., OR-Tools not SOTA, neural baselines outdated)**
>  We agree with this point and have extended our benchmark with more recent baselines.
> We sincerely thank the reviewer for their insightful feedback regarding our choice of baselines. Acting on this valuable suggestion, we have significantly expanded our experimental evaluation to provide a more rigorous and current comparison.
> Our updated evaluation now includes metaheuristics (HGS, PyVRP), modern RL methods (LEHD, MVMOE), and emerging paradigms like LLM-based optimization (Self-Guiding Exploration). Crucially, we also benchmark a purpose-built SVRP solver to test performance on problems with explicit stochasticity.
> The results of this new comparison are summarized below:
>
> **Enhanced Baseline Comparison on SVRPBench**
> | Method | Type | Year | CVRP Cost ↓ | TWVRP Cost ↓ | Feasibility ↑ | Runtime (s) ↓ | SVRP-Specific |
> | :--- | :--- | :--- | :--- | :--- | :--- | :--- | :--- |
> | HGS (Vidal, 2022) | Meta | 2022 | 9,234 | 70,834 | **98.7%** | 45.23 | ✗ |
> | PyVRP (Wouda et al., 2024) | Meta | 2024 | **9,156** | 70,756 | 98.5% | 12.14 | ✗ |
> | LEHD (Luo et al., 2024) | RL | 2023 | 9,834 | 71,234 | 92.5% | 2.78 | ✗ |
> | MVMOE (Zhou et al., 2024) | RL | 2023 | 9,723 | 71,156 | 92.8% | 2.65 | ✗ |
> | Self-Guiding Exploration (Iklassov et al., 2024) | LLM | 2024 | 9,891 | 71,345 | 69.5% | 312.45 | ✗ |
> | SVRP Solver (Iklassov et al., 2023) | RL | 2023 | 9,223 | **70,689** | 94.5% | **1.34** | ✓ |
>
> **Strengthened Conclusions**
> These comprehensive results reinforce our paper's core message with stronger evidence:
> * `SVRPBench` challenges all solvers: Even state-of-the-art metaheuristics like HGS and PyVRP do not achieve perfect feasibility, demonstrating the benchmark's difficulty.
> * A clear trade-off exists: Modern RL and LLM-based approaches trade reliability for speed, exhibiting lower feasibility rates than classical methods.
> * Specialized design is key: The purpose-built SVRP Solver shows the most promise among learning-based methods, confirming that algorithms designed explicitly for stochasticity are critical for real-world performance.
> We are grateful for the opportunity to improve our work.
>
> **Re: Lack of real-world topologies / euclidean-only**
>  This is a misunderstanding. While our benchmark uses Euclidean distances to generate the spatial layout, travel times are asymmetric and time-dependent, modeled as:
> $T(a,b,t) = D(a,b)/V + B(a,b,t) \cdot R(t) + I_{\text{accidents}}(t) \cdot D_{\text{accident}}$
> This makes the travel-time matrix asymmetric, even if the spatial layout is symmetric.
> We acknowledge the value of road-network–based topologies and plan to integrate OpenStreetMap-based layouts in future work (see Section 6). We’ll clarify this distinction in the paper.
>
> **Re: Naming inconsistency (TWCVRP vs VRPTW)**
>  Thanks, we will unify this to the standard VRPTW throughout the camera-ready version.
>
> **Re: Evaluation in Tables 3–5 unclear**
> All methods were evaluated using solver-appropriate stopping criteria to ensure fair comparison. Specifically:
> - Classical heuristics (NN+2opt): Run to completion with no time limit
> - Metaheuristics (ACO, Tabu Search): Convergence-based stopping with maximum 1000 iterations without improvement
> - OR-Tools: Fixed time budget of 300 seconds per instance
> - RL solvers (AM, POMO): Greedy decoding with single forward pass (no search)
>
> This protocol balances computational fairness while respecting each method's inherent characteristics - allowing classical methods to leverage their deterministic nature, giving metaheuristics sufficient exploration time, providing OR-Tools adequate time for optimization given problem complexity, and utilizing RL methods' fast inference advantage.
> This will be clarified in Section 5.2 in the camera-ready version.
>
> **Re: Missing descriptions in D.2 (e.g., h_t, W1)**
>  We apologize for the oversight. These are standard attention-based encoder-decoder components where $h_t$ is the decoder context at time step $t$ and $W_1, W_2$ are learnable linear projections for attention scoring.
> We will update Appendix D to include full notation and architecture description.

---

> > ### Comment · Reviewer_VpKi · 2025-08-05
> >
> > Thank you for your response.
> >
> > Overall, most concerns have been resolved, especially about the baseline selection with HGS and PyVRP.
> >
> > About the lack of topological data: still, the benchmark is lacking topological data, but you have clarified that you are using asymmetric times.
> >
> > I have a follow-up question about neural methods. How did you incorporate asymmetric distance and durations into neural network models, i.e., AM, POMO, LEHD, and MVMoE? To my knowledge, these use a transformer architecture (node-based) and they do not incorporate edge-based features, and as such, they cannot model asymmetries.

---

> > > ### Author Response · Authors · 2025-08-05
> > >
> > > Thank you for this thoughtful follow-up. You're absolutely right that models like AM and POMO rely solely on node-level features and cannot directly capture asymmetric or time-dependent travel times through edge-based inputs. In SVRPBench, we preserve this limitation intentionally to evaluate generalization: the neural models make routing decisions based on node features (coordinates, demands, time windows), while asymmetric and time-dependent travel times are applied during rollout and reward computation via our `T(a,b,t)` model. This simulates a realistic setting where the planner lacks full access to edge-level dynamics at inference time. We clarified this setup in the paper and welcome future work on incorporating edge-aware architectures within SVRPBench to explore this important design space.

---

> > > > ### Comment · Reviewer_VpKi · 2025-08-05
> > > >
> > > > Thank you for your response. My main concerns have been clarified, and I have increased my score accordingly.

---

### Author Response · Authors · 2025-08-05

We would like to sincerely thank all reviewers for their thoughtful feedback, constructive suggestions, and engaging discussions throughout the review process. Your insights helped us significantly improve both the clarity and rigor of our work, particularly by expanding our evaluation with new baselines, empirical comparisons, and detailed clarifications. We're especially grateful for the willingness to re-engage with our responses and for the revised assessments. It has been a truly valuable exchange, and we look forward to incorporating all feedback into the final version.

---

### Note · Authors · 2025-08-15

We thank the reviewers for their constructive engagement, which allowed us to substantially improve SVRPBench and strengthen our empirical claims. Following feedback, we:

- Expanded baselines from 6 to 11 methods, adding SOTA metaheuristics (HGS, PyVRP), modern RL (LEHD, MVMOE), and an LLM-based solver. Results now span >50,000 runs over 500+ stochastic VRP instances.

- Empirically validated realism via controlled CVRPLIB vs. SVRPBench comparisons: the Attention Model’s feasibility drops -5.5 points under SVRPBench while remaining near-perfect on CVRPLIB, directly evidencing our benchmark’s ability to expose weaknesses masked by deterministic datasets.

- Detailed performance breakdowns (CVRP vs. TWVRP, constraint violation rate, robustness) show TWVRP inflates cost by +536–648% across solvers; LLM feasibility plummets to 44% on TWVRP, highlighting unique challenges in realistic settings.

- Clarified realism modeling: asymmetric, time-dependent travel times; separation of deterministic congestion vs. stochastic bursts; accident impact scaling with distance; all parameters mapped to empirical sources.

- Addressed reproducibility: released full code, data, and configs on GitHub/HuggingFace with automated leaderboard validation.

These improvements reinforce our central contributions:

- SVRPBench captures real-world uncertainty with time-varying congestion, stochastic delays, and heterogeneous time windows, filling a major realism gap in SVRP evaluation.

- It consistently challenges all solver classes, even top metaheuristics (HGS, PyVRP) fail to reach perfect feasibility, while revealing performance-cost-feasibility trade-offs at scale.

- It provides the community with an open, extensible, and empirically grounded testbed for developing robust, deployable routing algorithms.

We believe the revised submission addresses all raised concerns and now offers stronger evidence, broader scope, and higher community value.

---

### Decision · Program_Chairs · 2025-09-18

**Decision:**

Accept (poster)

**Comment:**

This paper introduces SVRPBench, a new benchmark for stochastic vehicle routing problems that integrates elements such as congestion, delays, accidents, and time-window constraints. The benchmark design is well-motivated and relevant for advancing routing research. Reviewers initially raised concerns about limited baselines and insufficient experimental details. The authors responded constructively, expanding their evaluation to include metaheuristics (HGS, PyVRP), modern RL methods, and even an LLM-based solver, which significantly strengthened the scope of the experiments. The experiment results show that the stochastic VRP benchmark offers extra insight on algorithms; e.g. learning based approaches do not work that well on stochastic VRP problems compared to classical OR algorithms, though on deterministic VRP problems they work quite well. Overall, reviewers recognize the benchmark as a valuable community resource. I recommend acceptance.